# Cooperative assembly of p97 complexes involved in replication termination

Olga V. Kochenova[1,2], Sirisha Mukkavalli [3], Malavika Raman [3] & Johannes C. Walter [1,2] ✉

The p97 ATPase extracts polyubiquitylated proteins from diverse cellular structures in preparation for destruction by the proteasome. p97 functions with Ufd1-Npl4 and a variety of UBA-UBX co-factors, but how p97 complexes assemble on ubiquitylated substrates is unclear. To address this, we investigated how p97 disassembles the CMG helicase after it is ubiquitylated during replication termination. We show that p97[Ufd1-Npl4] recruitment to CMG requires the UBA-UBX protein Ubxn7, and conversely, stable Ubxn7 binding to CMG requires p97[Ufd1-Npl4]. This cooperative assembly involves interactions between Ubxn7, p97, Ufd1-Npl4, and ubiquitin. Another p97 co-factor, Faf1, partially compensates for the loss of Ubxn7. Surprisingly, p97[Ufd1-Npl4-Ubxn7] and p97[Ufd1-Npl4-Faf1] also assemble cooperatively on unanchored ubiquitin chains. We propose that cooperative and substrate-independent recognition of ubiquitin chains allows p97 to recognize an unlimited number of polyubiquitylated proteins while avoiding the formation of partial, inactive complexes.

p97 (Cdc48 in yeast) is a homohexameric, barrel-shaped molecular chaperone that uses ATP binding and hydrolysis to extract, remodel, and unfold polypeptide chains[1,2]. Substrate binding by p97 is achieved by a handful of p97 co-factors[3]. One of the best-studied co-factors, the Ufd1-Npl4 heterodimer, binds to polyubiquitin chains and couples p97/Cdc48 function to ubiquitin/proteasome-dependent protein degradation[4] (Supplementary Fig. 1a). Yeast Cdc48[Ufd1-Npl4] initiates substrate processing by unfolding one of the proximal ubiquitins using the MPN domain of Npl4[5]. Cdc48 ATP hydrolysis then translocates the substrate through its central pore. Thus, Ufd1-Npl4 is critical for recognition and unfolding of polyubiquitylated substrates.

Cells express additional proteins that are required to recruit p97 to substrates[6–8]. In mammals, five of these co-factors contain a ubiquitin regulatory X (UBX) domain that binds the N-terminal domain of p97/Cdc48 and a ubiquitin-associated (UBA) domain that binds ubiquitin chains (Supplementary Fig. 1a). Unlike p47, whose binding to p97 is mutually exclusive with Ufd1-Npl4[4], the other four UBA-UBX proteins (hUBXD7, hFAF1, hFAF2, and hSAKS1) can co-occupy p97 with Ufd1-Npl4[6]. hUBXD7 and hFAF1 bind stably to p97[Ufd1-Npl4] in vitro[9], suggesting that UBA-UBX proteins bind constitutively to p97.

However, other studies suggest their interaction with p97 in cells might be highly dynamic[6,10,11]. There is a growing list of proteins whose regulation by p97 depends on a specific UBA-UBX protein[6,12–20], and the number of clients is likely to be large[10,11,21,22]. Although a few examples exist where UBA-UBX proteins appear to bind a substrate[6,20], how UBA-UBX proteins generally contribute to substrate recognition is not understood. Several UBA-UBX proteins interact with specific E3 ubiquitin ligases, further tuning p97 function[6,23,24]. For example, the human UBA-UBX protein hUBXD7 binds to its substrate HIF1α and to neddylated CUL2, an E3 ubiquitin ligase. These interactions are detected in the absence of p97 (ref. 6), suggesting that hUBXD7 is the recognition module that connects p97[Ufd1-Npl4-UBXD7] to ubiquitylated HIF1α. In summary, little is known about how UBA-UBX co-factors work in a physiological context, including the dynamics of their association with p97[Ufd1-Npl4] complexes and most substrates. Additionally, it is unknown whether there is redundancy between any UBA-UBX proteins.

p97/Cdc48 regulates diverse chromatin-associated processes[2,25,26], including replisome disassembly[27–34]. Disassembly of terminated replisomes is initiated when the CMG (Cdc45, GINS, and the Mcm2-7

[1]Department of Biological Chemistry and Molecular Pharmacology, Harvard Medical School, Blavatnik Institute, Boston, MA 02115, USA. [2]Howard Hughes Medical Institute, Boston, MA, USA. [3]Department of Developmental Molecular and Chemical Biology, Tufts University School of Medicine, Boston, MA 02111, USA. ✉e-mail: johannes_walter@hms.harvard.edu

hexamer) helicase is ubiquitylated on its Mcm7 subunit by the cullin RING E3 ubiquitin CRL2$^{Lrr1}$ (SCF$^{Dia2}$ in yeast)[27-29,34-37]. The resulting K48-linked ubiquitin chain recruits p97/Cdc48, which removes CMG from chromatin[27,28,36,38]. In yeast, purified Cdc48$^{Ufd1-Npl4}$ is sufficient to promote CMG disassembly[39,40], and mutation of individual UBA-UBX co-factors does not affect CMG unloading[40]. In worms, UFD-1/NPL-4 and the UBA-UBX protein UBXN-3 are required for CUL-2$^{LRR-1}$-dependent CMG unloading[16,29]. However, the composition of the p97 complex that disassembles ubiquitylated vertebrate replisomes is unknown.

To understand how p97 complexes are assembled in a quasi-physiological setting, we used frog egg extracts, which recapitulate replication termination. We show that Ufd1-Npl4 and Ubxn7, the *Xenopus* ortholog of hUBXD7, are required for efficient CMG unloading. As expected, p97$^{Ufd1-Npl4}$ recruitment to CMG requires Ubxn7, but strikingly, the converse is also true: stable Ubxn7 binding to the CRL2$^{Lrr1}$-CMG complex depends on p97$^{Ufd1-Npl4}$. This cooperative p97 complex assembly on CMG involves multiple interactions between p97, Ufd1-Npl4, Ubxn7, CRL2$^{Lrr1}$, and ubiquitin. We also show that Faf1 (UBXN-3 in worms) can partially back-up Ubxn7 in this pathway. Importantly, the cooperative assembly of Ubxn7 and Faf1 with p97$^{Ufd1-Npl4}$ occurs on polyubiquitin chains independently of a substrate. Finally, we demonstrate that in mammalian cells, hUBXD7 promotes CMG unloading in S phase. Together, our results reveal how p97 complexes can assemble on a potentially unlimited number of substrates while avoiding the formation of partial, unproductive complexes.

## Results

### Ubxn7 recruits p97$^{Ufd1-Npl4}$ to CMG during replication termination

To explore how p97 promotes CMG unloading, we replicated DNA in *Xenopus laevis* egg extract, which supports a complete round of replication, including termination. As we reported previously, when DNA is replicated in the presence of the p97 inhibitor NMS-873 (p97i), CMG and CRL2$^{Lrr1}$ are greatly enriched on chromatin as measured in plasmid pull-downs (Fig. 1a, compare lanes 1 and 9; ref. 36). Under these conditions, p97, Ufd1, Npl4, and the UBA-UBX co-factor Ubxn7 are also enriched on chromatin, but not if fork convergence or CRL2$^{Lrr1}$ activity are blocked (Fig. 1a, compare lanes 1 and 9; ref. 36). These data suggested that Ubxn7 might cooperate with p97$^{Ufd1-Npl4}$ to promote CMG disassembly. To test this idea, we depleted Ubxn7 from egg extracts. In mock-depleted extract, replisome disassembly was complete 20 minutes after the initiation of replication, as seen by the absence of CMG subunits Mcm6, Cdc45, and Sld5 (a component of GINS) on chromatin (Fig. 1a, lane 1, Supplementary Fig. 1b, lane 4). In Ubxn7-depleted egg extract (Supplementary Fig. 1c, lane 1), CMG unloading was impaired, and polyubiquitylated Mcm7 remained on chromatin after 20 min (Fig. 1a, lane 2; Supplementary Fig. 1b). In the presence of p97i, depletion of Ubxn7 greatly reduced p97, Ufd1, and Npl4 enrichment on chromatin (Fig. 1a, compare lanes 9 and 10), but CRL2$^{Lrr1}$ and CMG were not affected (Supplementary Fig. 1e). Importantly, wild-type recombinant Ubxn7 protein (Fig. 1b) rescued CMG unloading and p97$^{Ufd1-Npl4}$ recruitment (Fig. 1a, lanes 3 and 11). Thus, Ubxn7 is required for efficient CMG unloading during replication termination due to a role in recruiting p97$^{Ufd1-Npl4}$ to chromatin.

### Efficient CMG unloading depends on Ubxn7 binding to ubiquitin, CRL2$^{Lrr1}$, and p97

Human Ubxn7 (hUBXD7) uses UBA, UIM (Ubiquitin Interacting Motif), and UBX domains to bind ubiquitin chains, neddylated CRL complexes, and p97, respectively (Fig. 1c), but the role of these interactions in processing hUBXD7 substrates is unclear[6,14,15,23,24,41]. To address this question, we first changed structurally-conserved hydrophobic residues in the UBA domain to alanines (Fig. 1c) and confirmed that the resulting Ubxn7$^{UBA*}$ protein was deficient in ubiquitin binding (Supplementary Fig. 1f, lane 6 vs. 7), as seen for the equivalent mutations in

the UBA domains of p47[42]. Surprisingly, both Ubxn7$^{UBA*}$ and Ubxn7 lacking the entire UBA domain (Ubxn7$^{ΔUBA}$) substantially rescued CMG unloading and p97 recruitment (Fig. 1a, lanes 4, 5, 12, 13), although Ubxn7$^{ΔUBA}$ was less proficient in p97 recruitment than Ubxn7$^{UBA*}$. Whereas purified Ubxn7$^{UBA*}$ barely co-precipitated K48 diubiquitin in vitro (Supplementary Fig. 1f), it still recovered some polyubiquitin chains from non-replicating egg extracts, albeit less efficiently than wild-type Ubxn7 (Supplementary Fig. 1g, lanes 23, 25, 37, and 39). Together, these data suggest that, in the absence of the UBA-ubiquitin interaction, Ubxn7 still recruits p97 to terminated CMG, perhaps by binding to Nedd8 on CRL2 via the UIM motif (Fig. 1d, e).

To test this idea, we purified Ubxn7$^{UIM*}$ (a serine to alanine substitution modeled on the UIM mutant of hUBXD7[24]), and showed that it was deficient for binding neddylated CRL2 in egg extracts (Supplementary Fig. 1g, compare lanes 26 and 23; note that neddylated CRL2 is probably de-neddylated during the FLAG-IP). Like Ubxn7$^{ΔUBA}$, Ubxn7$^{UIM*}$ partially rescued CMG unloading (Fig. 1a, lane 6), and it was less proficient than Ubxn7$^{ΔUBA}$ in p97 recruitment (Fig. 1a, lane 12 vs. 14). These data suggest that in the absence of a functional UIM domain, Ubxn7 can still function in p97$^{Ufd1-Npl4}$ recruitment (Fig. 1f), albeit less efficiently than in the absence of a UBA domain (Fig. 1e). To determine whether UBA and UIM domains are partially redundant, we examined Ubxn7$^{UBA*/UIM*}$ (Fig. 1b). This double mutant still co-IP'd p97, but it was less proficient in recovery of polyubiquitin chains from egg extracts than the single mutants (Supplementary Fig. 1g, lanes 39–41). Similarly, Ubxn7$^{UBA*/UIM*}$ was less proficient than the single mutants in supporting CMG unloading and p97$^{Ufd1-Npl4}$ recruitment (Fig. 1a, lanes 7 and 15; Fig. 1g). The effects of different Ubxn7 mutations on CMG unloading and p97 recruitment generally correlated with Ubxn7 levels on chromatin (Fig. 1a, lanes 11–15). We conclude that efficient CMG unloading by Ubxn7 involves partially redundant interactions with polyubiquitin chains on CMG and Nedd8 on CRL2$^{Lrr1}$, with the latter playing a somewhat greater role.

Finally, we created Ubxn7$^{UBX*}$ (Fig. 1b), which reduced co-immunoprecipitation of the p97$^{Ufd1-Npl4}$ complex with recombinant Ubxn7 almost to background levels in egg extracts (Supplementary Fig. 1g, compare lanes 28 and 22), as seen for the analogous mutation in hUBXD7[24], but had no effect on ubiquitin binding (Supplementary Fig. 1f, lane 8). Ubxn7$^{UBX*}$ was almost completely deficient for CMG unloading and p97$^{Ufd1-Npl4}$ recruitment (Fig. 1a, lanes 8 and 16; Fig. 1h). Surprisingly, despite retention of its ubiquitin- and Nedd8-interacting domains, Ubxn7$^{UBX*}$ was not enriched on chromatin above the background level (Fig. 1a, lane 16 vs. 8). Collectively, these data indicate that the recruitment of p97$^{Ufd1-Npl4}$ to ubiquitylated CMG requires a physical interaction between p97 and Ubxn7, which is recruited to the complex of neddylated CRL2$^{Lrr1}$ and ubiquitylated CMG via its UBA and UIM motifs (Fig. 1d). Moreover, the failure of Ubxn7$^{UBX*}$ to be recruited suggested the possibility that Ubxn7 retention on chromatin also requires a physical interaction with p97$^{Ufd1-Npl4}$.

### p97$^{Ufd1-Npl4}$ is required for stable interaction of Ubxn7 with ubiquitylated CMG

To address directly whether Ubxn7 binding to polyubiquitylated CMG requires p97$^{Ufd1-Npl4}$, we depleted Npl4 from egg extracts, which co-depleted endogenous Ufd1, as expected[43], but not p97 or Ubxn7 (Supplementary Fig. 2a, lane 1). Ubiquitylated CMG was retained on chromatin in Npl4-depleted extract (Fig. 2a, lanes 3 and 4), and supplementation of this extract with recombinant wild-type Ufd1-Npl4 (Fig. 2b and Supplementary Fig. 2a) restored CMG unloading (Fig. 2a, lanes 5 and 6). Interestingly, Ufd1-Npl4 depletion did not completely block CMG unloading (Fig. 2a, lanes 3 and 4) suggesting that either depletion was incomplete or that other p97 co-factor(s) can support inefficient CMG unloading. Strikingly, in the presence of p97i, the depletion of Ufd1-Npl4 abolished p97 and Ubxn7 recruitment to chromatin (Fig. 2c, lanes 3 and 4), and these defects were rescued by

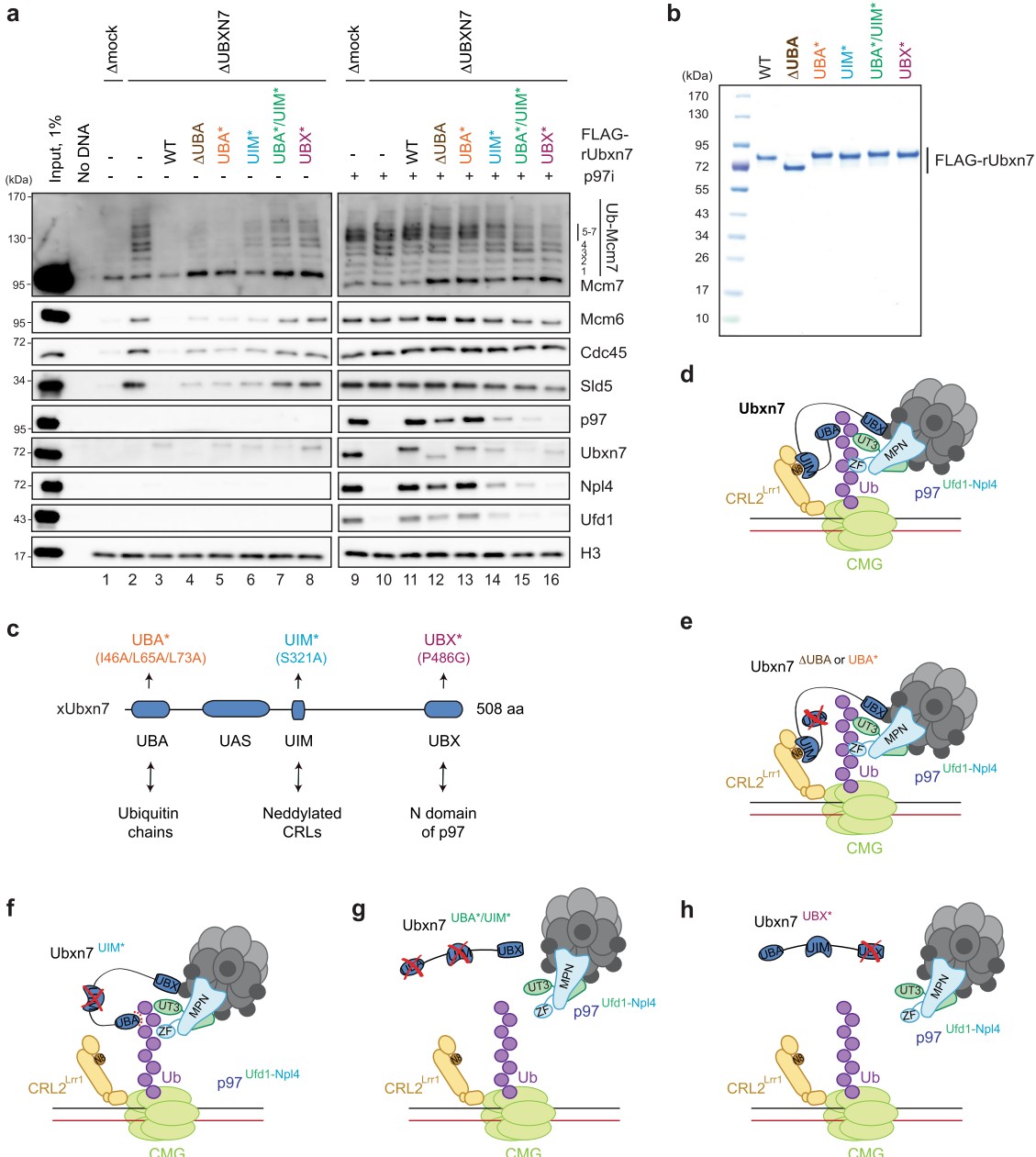

**Fig. 1 | Ubxn7 promotes unloading of terminated CMG in interphase. a** Analysis of chromatin-bound proteins in mock- or Ubxn7-depleted extracts in the absence or presence of p97i. Plasmid was incubated in total egg lysate to carry out licensing, and then exposed to nucleoplasmic extract (NPE) to initiate replication. Extracts were supplemented with rUbxn7 variants as indicated. At 20 min after NPE addition, plasmids were recovered with LacR-coated beads and immunoblotted for the indicated proteins. Ub-Mcm7, ubiquitylated Mcm7; Ubxn7$^{ΔUBA}$, deletion of the 1-74 N-terminal amino acids of Ubxn7; Ubxn7$^{UBA*}$, I46A/L65A/L73A amino acid substitutions in the UBA domain; Ubxn7$^{UIM*}$, S321A amino acid substitution in the UIM domain; Ubxn7$^{UBA*/UIM*}$, combined I46A/L65A/L73A and S321A amino acid substitutions; Ubxn7$^{UBX*}$, P486G amino acid substitution in the UBX domain. See panel (**c**) for Ubxn7 domain composition and Supplementary Fig. 1d for immunodepletion efficiency. Lanes 1-8 and 9-16 are parts of the same experiment and were analyzed on separate Western blots processed and imaged in parallel under identical conditions. **b** Coomassie-stained SDS PAGE gel of recombinant wild-type (WT) and mutant variants of Ubxn7. Abbreviations as in **a**. **c** Domain composition and known protein-protein interactions of *Xenopus* Ubxn7. UBX ubiquitin regulatory X domain, UIM Ubiquitin Interacting Motif, UBA ubiquitin-associated domain, UAS ubiquitin associating domain. Other abbreviations as in **a**. **d** A model for the Ubxn7-mediated p97 recruitment to ubiquitylated CMG during interphase termination. **e** A model for p97 recruitment to ubiquitylated CMG by UBA mutants of Ubxn7. **f** A model for p97 recruitment to ubiquitylated CMG by the UIM mutant of Ubxn7. **g** A model for p97 recruitment to ubiquitylated CMG the double UBA and UIM mutant of Ubxn7. **h** A model for p97 recruitment to ubiquitylated CMG by the UBX mutant of Ubxn7. In **d**–**h**, Ub ubiquitin, ZF NZF domain of Npl4, N8 Nedd8, red dotted lines indicate weak protein–protein interactions. Source data are provided as a Source Data file.

recombinant Ufd1-Npl4 (Fig. 2c, lanes 5 and 6). This result suggested that Ubxn7 binding to chromatin requires p97$^{Ufd1-Npl4}$. To rule out that extensive washing of chromatin in plasmid pull-downs disrupted Ubxn7 binding in the absence of Ufd1-Npl4, we performed chromatin immunoprecipitation (ChIP). Consistent with plasmid pull-down results, in extracts depleted of Ufd1-Npl4, Ubxn7 ChIP decreased to

the level observed when CMG ubiquitylation was blocked using MLN4924, a general inhibitor of cullin RING E3 ubiquitin ligase neddylation (Culi) (Supplementary Fig. 2bi–ii). Ubxn7 ChIP was rescued by recombinant Ufd1-Npl4 (Supplementary Fig. 2biii). As expected, Mcm7 binding was equivalent in all conditions, and, despite the presence of p97i, it dropped over time due to passive displacement of dormant

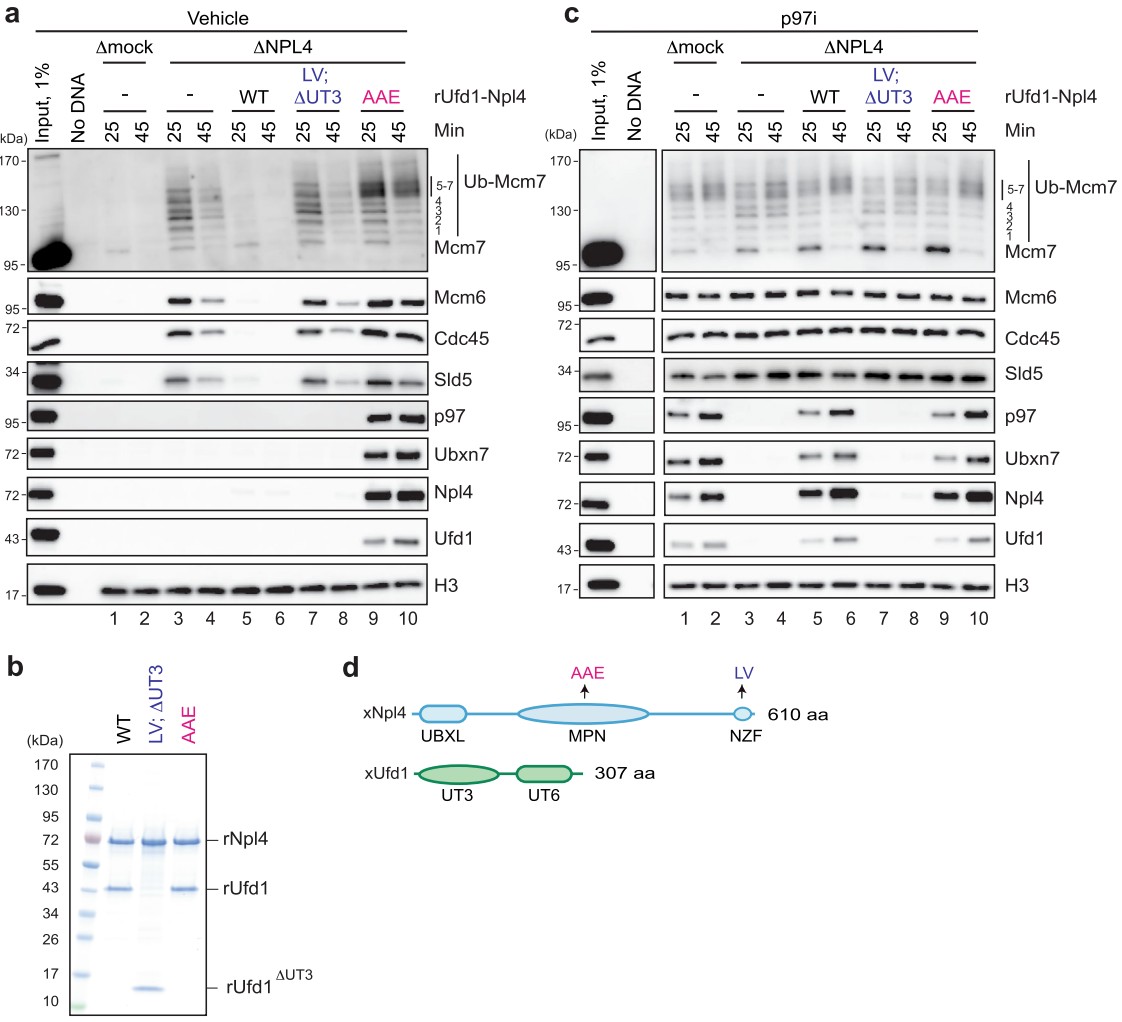

**Fig. 2 | The p97^Ufd1-Npl4 complex initiates CMG unloading in interphase. a** Analysis of chromatin-bound proteins in mock- or Npl4-depleted egg extracts in the absence p97i. Extracts were supplemented with recombinant wild-type or Ufd1-Npl4 mutants as indicated. At 25 and 45 min after replication initiation, plasmid DNA was recovered with LacR-coated beads and immunoblotted for the indicated proteins. ΔUT3, deletion of 2-215 amino acids of Ufd1. Ub-Mcm7, ubiquitylated Mcm7. Other abbreviations as in **d**. See also Supplementary Fig. 2a for mock and Ufd1-Npl4 immunodepletion efficiency. **b** An image of polyacrylamide gel of recombinant wild-type and mutant variants of Ufd1-Npl4. Abbreviations as in **d**. **c** As in **a**, but extracts were supplemented with p97i. See also Supplementary Fig. 2c for mock

and Ufd1-Npl4 immunodepletion efficiency. "Input, 1%", "No DNA" lanes, and lanes 1–10 are parts of the same experiment that were analyzed on separate Western blots processed and imaged in parallel under identical conditions. For Sld5, the contrast of the whole image was adjusted using ImageJ to enhance visibility of the bands. **d** Domain composition of *Xenopus* Npl4 and Ufd1. UBXL, ubiquitin regulatory X-like domain; MPN, Mpr1, Pad1 N-terminal domain; NZF, NPL4 Zinc Finger domain; UT3, Ufd1 truncation 3 domain; UT6, Ufd1 truncation 6 domain. LV, T592L/F593V amino acid substitutions in NZF; AAE, L240A/W243A/R244E amino acid substitutions in MPN. Source data are provided as a Source Data file.

Mcm2-7 complexes by replication (Supplementary Fig. 2bi–iii)[36,44]. These results demonstrate that CMG unloading requires Ufd1-Npl4, and that the binding of Ubxn7 requires p97^Ufd1-Npl4. Together with the demonstration that p97^Ufd1-Npl4 binding requires Ubxn7 (Fig. 1), we conclude that Ubxn7 and p97^Ufd1-Npl4 bind cooperatively to ubiquitylated CMG.

### Ubiquitin binding by Ufd1-Npl4 is critical for Ubxn7 recruitment to chromatin

We next tested whether ubiquitin binding by Ufd1-Npl4 is required for Ubxn7 recruitment and CMG disassembly. In higher eukaryotes, Npl4 binds polyubiquitin chains with its C-terminal $Zn^{2+}$-finger domain (NZF), and Ufd1 interacts with ubiquitin chains via its UT3 domain (Fig. 2d)[45–47]. To eliminate ubiquitin binding by *Xenopus* Ufd1-Npl4, we deleted the UT3 domain of Ufd1, and we introduced T592L/F593V (LV) amino acid substitutions into the NZF domain of Npl4, which effectively eliminates ubiquitin binding by human NPL4's NZF domain[47]. In non-replicating egg extracts, Ufd1^ΔUT3-Npl4^LV (Fig. 2b)

co-immunoprecipitated p97 normally while failing to bind polyubiquitin chains (Supplementary Fig. 2d, compare lanes 9 and 10). Importantly, Ufd1^ΔUT3-Npl4^LV did not support CMG disassembly (Fig. 2a, lanes 7 and 8) or p97 and Ubxn7 recruitment (Fig. 2c, lanes 7 and 8), suggesting that Ufd1-Npl4 binding to ubiquitin chains is essential for p97-dependent CMG unloading and for stable interaction of Ubxn7 with a polyubiquitylated substrate.

### Npl4's ubiquitin binding groove is critical for CMG unloading

In yeast, Npl4's MPN domain initiates processing of ubiquitylated substrates by unfolding the N-terminus of a substrate-proximal ubiquitin[5]. To test whether the MPN domain of *Xenopus* Npl4 is required for CMG unloading, we mutated three conserved amino acids (Fig. 2d; Supplementary Fig. 2e) within the MPN groove that stimulates ubiquitin unfolding by yeast p97^Ufd1-Npl4 (ref. 5). The resulting Ufd1-Npl4^AAE protein (Fig. 2b) retains its interaction with polyubiquitin chains (Supplementary Fig. 2d, lane 11), but is expected to lose ubiquitin unfolding activity. Strikingly, Ufd1-Npl4^AAE failed to support any

CMG unloading, and it trapped the p97[Ufd1-Npl4-Ubxn7] complex on chromatin, even in the absence of p97i (Fig. 2a, lanes 9 and 10), phenocopying the effects of p97 ATPase mutants[48]. By analogy with yeast, these results suggest that *Xenopus* Ufd1-Npl4 promotes CMG unloading by binding and unfolding the ubiquitin chain on CMG.

## Ubiquitin chains mediate complex formation between Ubxn7 and p97[Ufd1-Npl4]

Given our finding that Ubxn7 and p97[Ufd1-Npl4] bind cooperatively to chromatin during replication termination, it was surprising that these proteins co-IPed from extracts in the absence of DNA (Supplementary Fig. 1g). We hypothesized that Ubxn7 and p97[Ufd1-Npl4] assemble cooperatively on ubiquitin chains that are either tethered to endogenous substrates of unknown identity or not attached to any protein (Fig. 3ai; "free Ub chains"). Consistent with this idea, Ubxn7 co-IP'd not only Npl4, but also long ubiquitin chains (Supplementary Fig. 1g, lanes 23 and 37), and Npl4 co-IP'd Ubxn7 and ubiquitin chains (Supplementary Fig. 2d, lanes 9 and 14). Moreover, Ubxn7[UBX*] that is unable to interact with p97 failed to efficiently recover polyubiquitin chains (Supplementary Fig. 1g, lanes 42 vs. 37; Fig. 3aii), and the same was true when Ubxn7[WT] was precipitated from extracts depleted of Ufd1-Npl4 (Fig. 3b, lanes 18 vs. 24). Conversely, the ubiquitin binding-deficient Ufd1[ΔUT3]-Npl4[LV] mutant not only failed to recover ubiquitylated substrates, it also did not bind Ubxn7 or another UBA-UBX protein, Faf1 (Supplementary Fig. 2d, lanes 10 and 15; Fig. 3aiii), which we show below backs up Ubxn7 for CMG unloading. In contrast, Ufd1-Npl4[AAE] that trapped p97 on ubiquitylated CMG exhibited an enhanced Ubxn7 and Faf1 recovery from egg extracts in the absence of p97i (Supplementary Fig. 2d, lanes 8 vs 11). Finally, p97[Ufd1-Npl4] co-precipitated Ubxn7[ΔUBA] much less strongly than Ubxn7[WT] (Supplementary Fig. 3b, lanes 12 vs. 10), and the residual recovery of Ubxn7[ΔUBA] was dependent on Ubxn7's interaction with neddylated cullins (Supplementary Fig. 3b, lane 14). In summary, even in the absence of DNA, Ubxn7 and p97[Ufd1-Npl4] bind cooperatively to anonymous substrates and/or free ubiquitin chains. This assembly requires Ubxn7's ubiquitin and p97 binding domains, as well as Ufd1-Npl4's ubiquitin binding domains, suggesting that ubiquitin chains are critical to scaffold the assembly.

To directly examine the role of ubiquitin chains in p97 complex assembly, we inhibited de novo ubiquitylation in egg extracts using the E1 enzyme inhibitor TAK-243 (E1i). We also added the deubiquitylating enzyme (DUB), Usp2, to degrade pre-existing polyubiquitin chains (Fig. 3c, lane 4). Consistent with a critical role for ubiquitin chains, Ubxn7 and Faf1 failed to co-IP with Ufd1-Npl4 in the absence of endogenous ubiquitin chains (Fig. 3c, lane 13 vs. 19; Fig. 3aiv). Supplementation of these extracts with unanchored, recombinant K48-linked ubiquitin chains containing 2-7 ubiquitins (K48 Ub$_{(2-7)}$) (Fig. 3c, lane 5; a Usp2 inhibitor was added to extracts to preserve the chains) restored complex formation between p97[Ufd1-Npl4] and Ubxn7/Faf1 (Fig. 3c, lane 22). Interestingly, purified p97[Ufd1-Npl4] and Ubxn7 interacted independently of ubiquitin chains in buffer, even at very low protein concentrations (Supplementary Fig. 4a–c), as seen previously[9]. Together, these results show that in the complex environment of a cellular lysate, Ubxn7 and p97[Ufd1-Npl4] only assemble in the presence of ubiquitin chains, including ones that are not attached to a substrate.

We also explored the interaction of Ubxn7 with Cul2, and how this interaction contributes to p97 complex assembly on ubiquitin chains. Although mutating the UBX domain dramatically reduced Ubxn7's recovery of polyubiquitin chains, its effect on recovery of neddylated Cul2 was modest (Supplementary Fig. 1g, lanes 42 vs. 37). Similarly, in the absence of Ufd1-Npl4, when Ubxn7 binds inefficiently to p97 and ubiquitin chains, Ubxn7 still retained some Cul2 binding that depended on Cul2 neddylation (Fig. 3b, compare lanes 18, 22, and 24). However, when we raised the salt concentration to 100 mM, the Ubxn7-Cul2 interaction seen in the absence of Ufd1-Npl4 was abolished (Fig. 3b, lane 36). This result indicates that in the absence of full

complex assembly, Ubxn7 binds weakly to neddylated CRL2 complexes. In undepleted extract, Culi decreased association of endogenous Ubxn7 with Ufd1-Npl4 (Fig. 3c, lanes 13 vs. 16) and the UIM mutant of Ubxn7 was inefficiently recruited to chromatin during termination (Fig. 1a), consistent with the Ubxn7-CRL2 interaction contributing to p97 complex formation. Surprisingly, the interaction of p97[Ufd1-Npl4] with Faf1, which lacks a UIM domain, was also inhibited by Culi (Fig. 3c, lanes 13 vs. 16). Given that Faf1 co-IPs with CRL1 and CRL3 complexes[6,19,22], this suggests that p97[Ufd1-Npl4-Faf1] might assemble on ubiquitin chains made by these E3 ligases. The residual complex formation between Ubxn7 and Faf1 with p97[Ufd1-Npl4] in the presence of Culi probably occurs through assembly on ubiquitin chains made by non-cullin RING ubiquitin ligases[6]. Together, our findings indicate that in general, assembly of p97[Ufd1-Npl4-UBA-UBX] on ubiquitin chains requires the interaction of Ufd1-Npl4 and the UBA domain with ubiquitin, as well as UBX domain binding to p97 (Fig. 3ai). Complex formation is augmented by Cullin neddylation. Furthermore, our observations argue that the cooperative assembly of Ubxn7 and Faf1 with p97[Ufd1-Npl4] on ubiquitin chains does not require direct interaction with a substrate.

## Faf1 helps to unload CMG in the absence of Ubxn7

Although Ubxn7 depletion delayed CMG unloading, the helicase was eventually removed from DNA in this setting (Fig. 4a, lanes 4–6 and Supplementary Fig. 1b). In worms, the UBA-UBX protein, UBXN-3, promotes CMG unloading[16,29]. Therefore, we asked whether Faf1, the *Xenopus* ortholog of UBXN-3, unloads CMG in Ubxn7-depleted egg extracts. Immunodepletion of Faf1 alone (Supplementary Fig. 5a) did not affect CMG unloading (Fig. 4a, lanes 1–3 vs. 7–9). However, when Ubxn7 and Faf1 were co-immunodepleted, replisome unloading was further delayed compared to Ubxn7-depleted egg extract (Fig. 4a, lanes 10–12 vs. 4–6). rUbxn7 fully restored CMG unloading in Ubxn7/Faf1-depleted extract (Fig. 4a, lanes 13–15), while recombinant Faf1 (rFaf1) (Fig. 4b, lane 1) rescued CMG unloading roughly to the level observed in the Ubxn7-depleted extract (Fig. 4a, lanes 16–18 vs. 4–6; see also Supplementary Fig. 5b). These data indicate that Faf1 can partially compensate for the loss of Ubxn7. The fact that double Ubxn7 and Faf1 depletion did not completely block CMG unloading (Fig. 4a, lanes 10–12) suggests that depletion was incomplete or that still other p97 co-factor(s) or p97[Ufd1-Npl4] alone support inefficient CMG unloading.

## Portable enhancement of CMG unloading by the UIM motif

We wanted to know why Faf1 is less active than Ubxn7 in promoting CMG unloading. We first tested the possibility that Ubxn7 binds better than Faf1 to the relatively short K48-linked ubiquitin chains that are formed on CMG by CRL2[Lrr1] (e.g. Fig. 1a and[28]). However, we found that Ubxn7 and Faf1 both exhibited preferential binding to K48-linked chains (Supplementary Fig. 5c, lanes 19–34), and Faf1 bound short K48 chains at least as well as Ubxn7 (Fig. 4c). Thus, intrinsic differences in ubiquitin binding do not explain the different activities of Ubxn7 and Faf1. An alternative idea was that binding of Ubxn7's UIM domain to neddylated CRL2[Lrr1] renders Ubxn7 more active for CMG unloading than Faf1, which lacks a UIM domain. This model is consistent with the observation that Ubxn7[UIM*] was deficient in p97[Ufd1-Npl4] recruitment and CMG unloading (Fig. 1a). To further test this idea, we created a Faf1 chimera containing a 22-amino acid segment that encompasses the predicted α-helix portion of Ubxn7's UIM motif (Faf1[UIM]) (Fig. 4b, d)[24,49,50]. Faf1[UIM] was more active than Faf1[WT] in promoting CMG unloading in Ubxn7/Faf1-depleted extracts (Fig. 4a, lanes 16–21). When added at endogenous levels, Faf1[UIM] was not as active as endogenous Ubxn7 (Fig. 4a, compare lanes 19–21 and 13–15), but when added in excess, Faf1[UIM] activity was comparable to that of endogenous Ubxn7 (Supplementary Fig. 5b; compare lanes 19–21 and 13–15). Together, these results argue that Faf1 is less active than Ubxn7, in part due to the lack of a UIM domain.

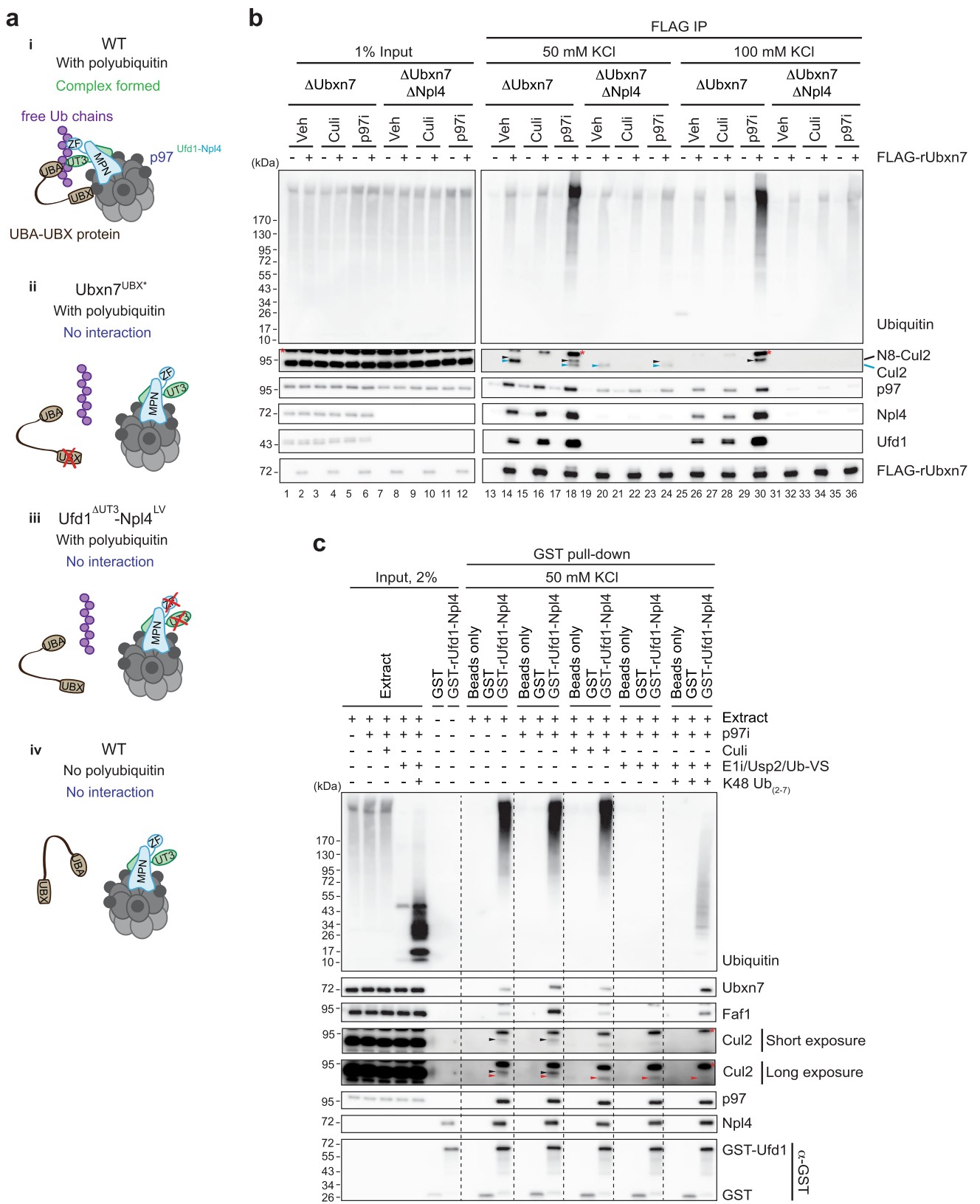

Unexpectedly, despite there being substantial CMG unloading in Ubxn7-depleted extract, much of which involves Faf1 (Fig. 4a), we could barely detect any p97 recruitment in Ubxn7-depleted extracts (Fig. 1a, lane 10 and Fig. 4e. lane 2). This was also true when Ubxn7/Faf1-depleted extracts were supplemented with Faf1^WT (Fig. 4a, lane 6). Although Faf1^UIM exhibited better neddylated Cul2 binding than Faf1^WT

(Supplementary Fig. 6b, lanes 15–16) and was recovered more efficiently than Faf1^WT in plasmid pull-down (Fig. 4e, lanes 6–7), it was still less effective than Ubxn7 in supporting p97 recruitment to ubiquitylated CMG (Fig. 4e, compare lanes 5 and 7). Altogether, these results show that Ubxn7's UIM domain is a portable motif that enhances the ability of UBA-UBX proteins to promote CMG unloading, even though

**Fig. 3 | Ubiquitin chains mediate interaction between UBA-UBX proteins and p97$^{Ufd1-Npl4}$. a** A model for ubiquitin chain-mediated complex assembly between p97$^{Ufd1-Npl4}$ and UBA-UBX proteins. **b** Co-immunoprecipitates of FLAG-rUbxn7 from Ubxn7- or Ubxn7/Npl4-depleted egg extracts in the presence or absence of p97i or Culi were analyzed with the indicated antibodies. FLAG co-immunoprecipitates were washed with 50 mM or 100 mM KCl in IP buffers. See Supplementary Fig. 3a for verification that Ubxn7 and Npl4 were depleted. Note: Cul2 undergoes de-neddylation during the FLAG immunoprecipitation procedure, and the degree of de-neddylation varies between extracts. De-neddylation is partially prevented by p97i (lanes 14 vs. 18), and further inhibited by 100 mM KCl (lane 30). Recovery of both neddylated and de-neddylated Cul2 is inhibited by Culi (lane 16), confirming that neddylation is required for its association with Ubxn7. Black arrowheads, neddylated Cul2; blue arrowheads, un-neddylated Cul2. Red asterisks, non-specific bands. N8-Cul2, neddylated Cul2. Lanes 1–12 and 13–36 are parts of the same experiment that were analyzed on separate Western blots processed and imaged in parallel under identical conditions. For Cul2 and Ufd1, the contrast of the whole image was adjusted using ImageJ to enhance visibility of the bands. **c** Co-immunoprecipitates of GST or wild type GST-Ufd1-Npl4 heterodimer from non-replicating egg extracts in the presence of the indicated additives were analyzed with indicated antibodies. Black arrowheads indicate neddylated Cul2 (E1i also blocks neddylating enzymes). Red arrowheads most likely indicate a cross-reacting band or non-specific recovery of unneddylated Cul2. For Cul2, the contrast of the whole image was adjusted using ImageJ to enhance visibility of the bands. Red asterisks, non-specific bands. See Supplementary Fig. 3c for verification of Npl4 depletion. Source data are provided as a Source Data file.

it is insufficient to support normal recovery of p97 complexes on chromatin (see Discussion).

## hUBXD7 is required for CMG unloading in mammalian cells

To determine whether hUBXD7 is required for CMG unloading in mammalian cells, we created a hUBXD7 knockout (KO) in HeLa Flp-in TRex (HFT) cells using CRISPR-Cas9 gene editing. To measure CMG unloading, we extracted asynchronous wild-type and hUBXD7 KO cells with high salt and non-ionic detergent to remove soluble proteins, stained cells with MCM7 antibody and propidium iodide, and subjected them to flow cytometry[51,52] (Supplementary Fig. 7a). As reported previously[52], chromatin-bound MCM7 declined as WT cells passed through S phase (Supplementary Fig. 7b, c). Notably, although MCM7 signal also declined throughout S phase in hUBXD7 KO cells, these cells retained more MCM7 on chromatin in G2 than wild-type cells (Fig. 5a). The amount of MCM7 retained was similar to that observed in the presence of the p97 inhibitor CB-5083, consistent with a substantial unloading defect in hUBXD7 KO cells. We also assessed CMG unloading via chromatin fractionation. Cells were synchronized at the G1/S boundary using double thymidine block and released into S phase for 10 h in the presence of the CDK1 inhibitor RO-3306 to arrest cells at the G2/M boundary[53], which prevents the mitotic pathway of CMG unloading[32]. Both wild-type and hUBXD7 KO cells progressed normally through the S phase after release as judged by flow cytometry and cyclin levels (Supplementary Fig. 7e, f). We then prepared soluble and chromatin-bound fractions of wild-type and hUBXD7 KO cells and blotted them for CMG components. In wild-type cells, MCM3, MCM7, and CDC45 dissociated from chromatin almost completely as cells passed through S phase, whereas a substantial fraction of these proteins was retained on chromatin in the hUBXD7 KO cells (Fig. 5b, compare lanes 3 and 6). Together, our results indicate that hUBXD7 is the primary UBA-UBX protein that cooperates with p97 for CMG unloading in mammalian cells.

## Discussion

In this study, we show that Ubxn7 is critical for CMG unloading during vertebrate replication termination, and we provide novel insights into how p97 complexes assemble with UBA-UBX proteins on ubiquitylated substrates. Our results show that in *Xenopus* egg extracts, Ubxn7 and p97$^{Ufd1-Npl4}$ assemble cooperatively on polyubiquitin chains. In support of this idea, we observe co-IP of Ubxn7, p97, Ufd1-Npl4, and ubiquitin chains. Formation of these p97$^{Ufd1-Npl4-Ubxn7}$-ubiquitin complexes requires Ubxn7's p97 (UBX), ubiquitin (UBA), and neddylated CRL (UIM) binding domains, as well as ubiquitin binding by Ufd1-Npl4, and complex assembly is abolished when ubiquitin chains are degraded in the extract. Consistent with this cooperative assembly occurring on a bona-fide substrate, CMG, the recruitment of p97 and Ufd1-Npl4 to chromatin requires Ubxn7's UBA, UIM, and UBX domains, and conversely, Ubxn7 recruitment requires Ufd1-Npl4's binding to ubiquitin. Thus, stable Ubxn7 and p97$^{Ufd1-Npl4}$ binding to ubiquitin chains involves multi-valent, cooperative interactions between Ubxn7, p97, Ufd1-Npl4,

ubiquitin, and CRL2$^{Lrr1}$ (Fig. 6b). Faf1 appears to behave similarly to Ubxn7, since it also co-IPs with p97$^{Ufd1-Npl4}$ dependent on the availability of and Ufd1-Npl4 binding to ubiquitin chains. Interestingly, purified hFAF1 and hUBXD7 bind to p97$^{Ufd1-Npl4}$ in the absence of ubiquitin chains (Ref. 9 and Supplementary Fig. 4), and Ufd1-Npl4 binds ubiquitin in the absence of other proteins[46]. Therefore, a mechanism must exist in cell lysates that restrains formation of higher order p97 assemblies. Assembly might be inhibited by a negative regulator, post-translational modifications (reviewed in ref. 54), or non-specific interactions with other macromolecules that drastically reduce the effective concentrations of p97, Ufd1-Npl4, and UBA-UBX proteins in cells[9,55].

Our evidence suggests that in the absence of full p97$^{Ufd1-Npl4-Ubxn7}$-ubiquitin complex assembly, a partial complex between Ubxn7 and CRL2$^{Lrr1}$ on ubiquitylated CMG and other substrates might form. Consistent with this model, Ubxn7$^{UBX*}$, which does not bind p97, co-IP'd weakly with neddylated Cul2 and polyubiquitylated proteins at 50 mM salt (Supplementary Fig. 1g, lane 28), and in Ufd1-Npl4-depleted extract, Ubxn7$^{WT}$ co-IP'd Cul2 in 50 mM but not 100 mM salt (Fig. 3b, lanes 24 vs 36; see also Legend). Finally, Ubxn7$^{UBX*}$ binding to terminating CMG was not detected in chromatin pull-downs at 100 mM salt (Fig. 1a, lane 16). Based on these results, we infer that in the absence of p97$^{Ufd1-Npl4}$, Ubxn7 binds weakly and transiently to ubiquitylated CMG-CRL2$^{Lrr1}$ at physiological salt concentrations. This weak binding might constitute the earliest event in CMG unloading, when ubiquitin chains are too short for recruitment of p97$^{Ufd1-Npl4}$ (Fig. 6a). Consistent with our observations, prior studies that reported ubiquitin-independent interaction between hUBXD7 and CRL2 involved hUBDX7 over-expression and sub-physiological salt concentrations (70 mM)[6,23,24].

Interestingly, mutations within Npl4's MPN domain that are predicted to prevent ubiquitin unfolding[5] trap the p97$^{Ufd1-Npl4-Ubxn7}$ complex on ubiquitylated CMG to the same extent as an allosteric p97 ATPase inhibitor (Fig. 2a, c). Therefore, in egg extracts, cooperative and multivalent p97$^{Ufd1-Npl4-Ubxn7}$ assembly appears to occur independently of ubiquitin entry into the p97 central channel. We propose that p97$^{Ufd1-Npl4-Ubxn7}$ provides the platform on which ubiquitin is unfolded (Fig. 6b), whereafter the Mcm7 subunit is threaded through the p97 channel (Fig. 6c).

An important question is whether cooperative assembly of the p97 complex on ubiquitin chains also occurs in cells. Consistent with this idea, binding of another UBA-UBX protein, hSAKS1, to p97 is stimulated by free polyubiquitin chains in vitro[56]. Intriguingly, in human cells, UBA-UBX protein and ubiquitin binding to p97 increases in the presence of p97i[10,11], which is not predicted if UBA-UBX proteins bind constitutively to p97. Another interpretation of this result is that p97i induces a conformational change in p97 that enhances binding to UBA-UBX proteins[10,11]. However, in egg extracts, p97i had no effect on p97 binding to Ubxn7 and Faf1 in the absence of ubiquitin chains (Fig. 3c, lane 19), arguing against this interpretation. Importantly, the p97i effect is also expected if UBA-UBX proteins bind stably to ubiquitin chains first, followed by p97 recruitment ("sequential binding"), so the

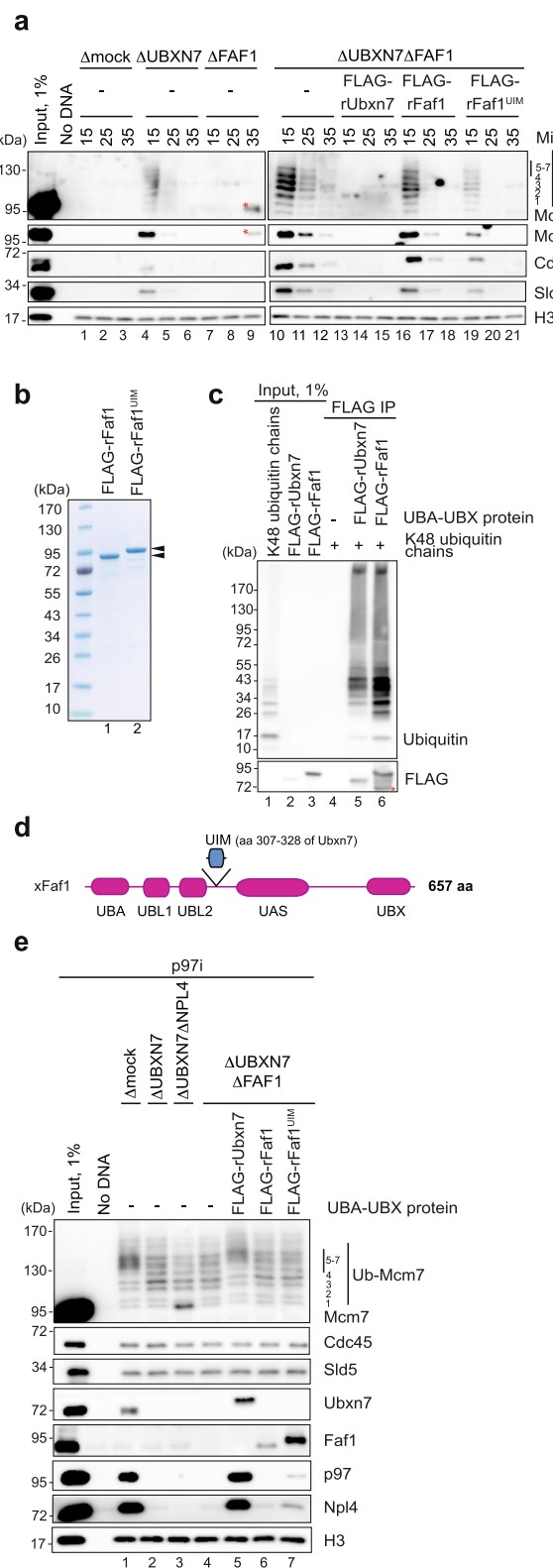

**a**

**b**

**c**

**d**

**e**

**Fig. 4 | Faf1's back-up function in CMG unloading can be enhanced by Ubxn7's UIM motif. a** Analysis of chromatin-bound proteins in mock-, Ubxn7-, Faf1-, or Ubxn7/Faf1-depleted extracts. At indicated time points after replication initiation, plasmid DNA was recovered with LacR-coated beads and immunoblotted for the indicated proteins. Lanes 1–9 and 10–21 are parts of the same experiment that were analyzed on separate Western blots processed and imaged in parallel under identical conditions. Red asterisks indicate non-specific band that is likely contamination with non-reduced heavy chains of rabbit IgG. See Supplementary Fig. 6a for immunodepletion efficiency and 6b for independent replicate. Ub-Mcm7, ubiquitylated Mcm7. **b** An image of polyacrylamide gel of recombinant wild-type Faf1 and Faf1[UIM] chimera. Black arrowheads indicate Faf1-specific bands. **c** Analysis of co-immunoprecipitates of FLAG-rUbxn7 or FLAG-rFaf1 with K48-linked ubiquitin chains. Recombinant Ubxn7 and Faf1 were detected by using anti-FLAG antibodies. Red asterisk indicates C-terminal truncation of recombinant Faf1. IP immunoprecipitation. **d** Domain composition of *Xenopus* Faf1 and the Faf1[UIM] chimera. In Faf1[UIM], the UIM motif of Ubxn7 was inserted between amino acid residues 293 and 294 of Faf1. UBX ubiquitin regulatory X domain, UIM Ubiquitin Interacting Motif, UBA ubiquitin-associated domain, UBL1, 2 ubiquitin like domain 1, 2, UAS ubiquitin associating domain. **e** Analysis of chromatin-bound proteins in mock-, Ubxn7-, Ubxn7-Npl4-, or Ubxn7/Faf1-depleted extracts in the presence of p97i and wild-type Ubxn7, wild-type Faf1, or the Faf1[UIM] chimera. See also Supplementary Fig. 7a for immunodepletion efficiency. Source data are provided as a Source Data file.

Ubxn7's and Faf1's recovery of most polyubiquitin chains depends on their cooperative assembly with p97[Ufd1-Npl4] on polyubiquitylated substrates. In summary, more work is required to test the idea that p97 complexes assemble cooperatively on ubiquitin chains in mammalian cells.

A related question is how UBA-UBX proteins recognize their substrates. It was reported that hUBXD7 binds to polyubiquitylated HIF1α independently of p97 (ref. [6], [24]). While these experiments involved overexpression of FLAG-hUBXD7 and thus may not reflect behavior of the endogenous protein, HIF1α might be a special case. We showed that the complex between Ubxn7 and Faf1 with p97[Ufd1-Npl4] can be formed on K48 ubiquitin chains that are not attached to a substrate (Fig. 3c). Given that there are only four classic UBA-UBX proteins and probably hundreds of p97 substrates[10,11,21], we propose that p97[Ufd1-Npl4-(UBA-UBX)] generally assembles on K48-linked ubiquitin chains without direct contact to the substrate. Consistent with our results, Rapoport and colleagues reported recently that in yeast, a similar spectrum of proteins is associated with Cdc48/p97 and K48-linked ubiquitin chains, suggesting that Cdc48/p97 recognizes primarily the ubiquitin chain and not the attached protein[58].

Cooperative p97[Ufd1-Npl4-(UBA-UBX)] assembly allows a relatively small pool of p97[Ufd1-Npl4] complexes to undergo dynamic assembly with UBA-UBX proteins on polyubiquitin chains, as needed. Cooperative assembly also avoids the formation of unproductive protein complexes, as proposed previously[9]. Thus, the fact that UBA-UBX proteins cannot bind stably to ubiquitin chains in the absence of p97[Ufd1-Npl4] ensures that UBA-UBX proteins do not bind chains unproductively, which might inhibit DUBs, the proteasome, or other ubiquitin-metabolizing proteins. Consistent with this idea, overexpression of UBA-UBX proteins or UBA domains alone causes defects in proteasome-mediated destruction of certain substrates[18,59]. Furthermore, to avoid unproductive binding of UBA-UBX proteins to p97, Ufd1-Npl4 is necessary for this interaction, even in reconstitution experiments where the requirement for ubiquitin chains is bypassed (Supplementary Fig. 4c, lane 3 and ref. [9]). Why p97[Ufd1-Npl4] binding to ubiquitin chains should be dependent on UBA-UBX proteins is an important question. We speculate that the cooperative multi-step assembly mechanism depicted in Fig. 6 is slow, but ultimately leads to very stable complexes that can initiate ubiquitin unfolding. Such slow assembly of the p97 complex might grant the proteasome priority access to ubiquitylated substrates. In this way, only proteins that cannot be processed by the proteasome would be acted on by p97.

p97i effect does not clearly distinguish between sequential and cooperative binding. Other data show that mutation of the UBX domain abolishes hUBXD7's and hFAF1's ubiquitin-binding in human cells[6,24,57], a phenomenon we also observed for *Xenopus* Ubxn7 in egg extracts (Supplementary Fig. 1g and Fig. 3b). While this phenomenon was previously explained by either a broad effect of these mutations on the overall protein structure[24] or the requirement for UBA-UBX proteins to pre-form a complex with p97[Ufd1-Npl4] prior to interaction with ubiquitylated substrates[57], it is also consistent with our model that

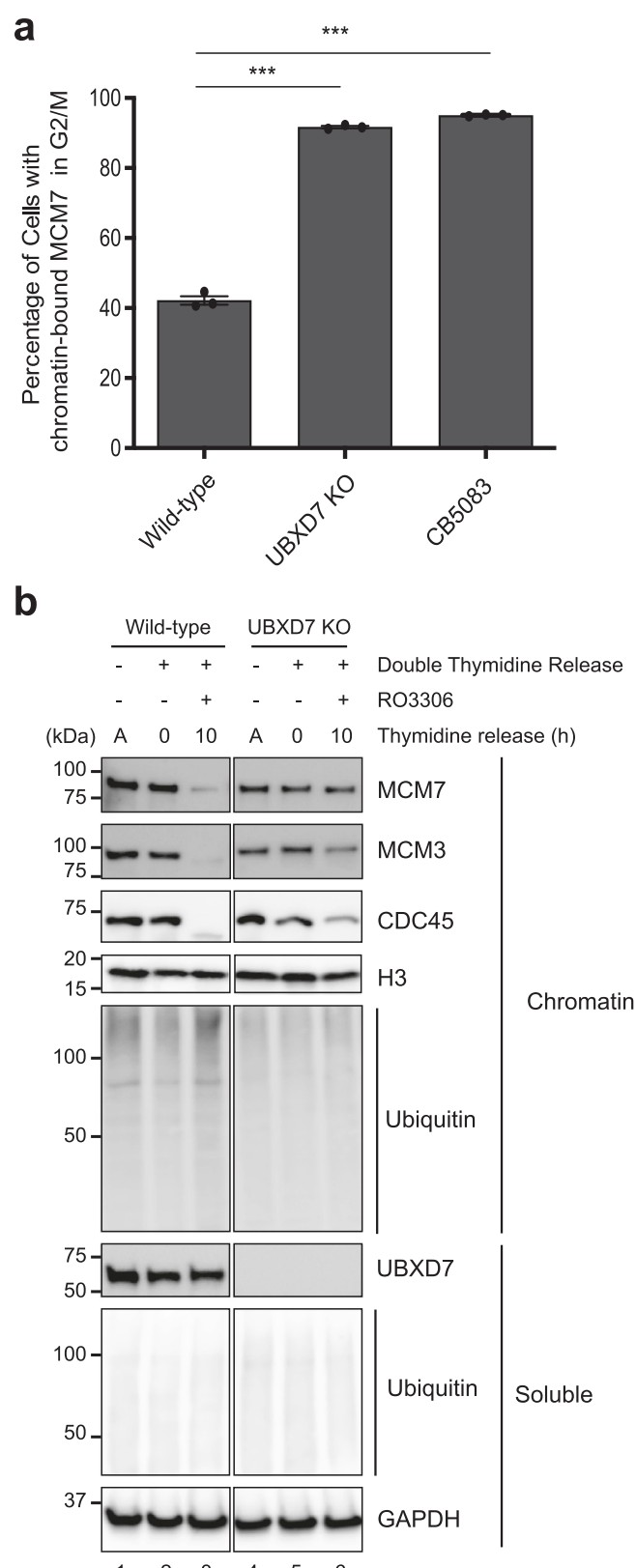

**Fig. 5 | hUBXD7 is required for CMG unloading in mammalian cells.**
**a** Quantification of MCM7 levels on chromatin in the G2/M phase of cell cycle in wild-type HFT cells in the presence or absence of the p97 inhibitor CB-5083 and in hUBXD7 KO cells. Bars represent mean, and error bars represent SD from three independent repeats. Asterisks indicate statistical significance compared with the control, as determined by one way ANOVA and Tukey test for multiple comparisons (***$P < 0.0001$). See also Supplementary Fig. 7. **b** Wild-type or hUBXD7 KO HFT cells were allowed to grow asynchronously or they were synchronized with double thymidine and then optionally released into RO3306 for 10 h, as indicated. Soluble and chromatin bound proteins were blotted for the indicated proteins. The images are all part of the same Western blot, which was cropped to remove irrelevant information between lanes 3–4. A, asynchronous. Source data are provided as a Source Data file.

efficient p97 recruitment than Ubxn7. Interestingly, in *C. elegans*, which lacks Ubxn7 orthologs[60], UBXN-3/Faf1 is the primary co-factor that promotes CMG unloading[16,32], suggesting that Ubxn7 supplanted this role of Faf1 in vertebrates. Future work will be required to fully understand how Faf1 differs from Ubxn7 and UBXN-3. Notably, even in extracts depleted of Ubxn7 and Faf1, CMG was ultimately still unloaded. This unloading might involve p97$^{Ufd1-Npl4}$ acting alone[61], or it could reflect the action of another UBA-UBX protein. Among the four vertebrate UBA-UBX co-factors, only Ubxn7 and Faf1 show a strong nuclear localization[20,22,24]. In contrast, Saks1 and Faf2 primarily operate in the cytoplasm and endoplasmic reticulum[17,22,56,62–65]. Because we perform replication in egg extracts containing a mixture of nuclear and cytoplasmic proteins[66], we speculate that in the absence of Ubxn7 and Faf1, Saks1 and/or Faf2 are available to unload CMG. Similarly, virtually all UBA-UBX co-factors interact with hyperubiquitylated HIF1α in human cell extracts, even though it is considered to be primarily a hUBXD7 substrate[6,24]. Thus, while mechanisms probably exist to prioritize the use of certain UBA-UBX proteins over others (e.g. contacts with the E3 ligase, subcellular localization), there is extensive redundancy among these p97 co-factors. These considerations underscore the concept that direct interactions between UBA-UBX proteins and substrates are unlikely to be essential for unfolding.

In summary, we have identified Ubxn7 as the primary UBA-UBX protein that promotes CMG unloading in vertebrates. Importantly, we have shown that UBA-UBX proteins assemble cooperatively with p97$^{Ufd1-Npl4}$ on ubiquitin chains, generating a stable complex that likely unfolds substrates without contacting them directly. This mechanism allows recognition of an unlimited number of targets while avoiding assembly of partial, inactive complexes. In agreement with our conclusions, the Labib and Gambus groups recently reported that Ubxn7 is the primary UBA-UBX protein that promotes CMG unloading in vertebrates[61,67], and that Faf1 can partially substitute for Ubxn7[61].

## Methods
### Statistics and reproducibility
All in vitro experiments were performed at least twice, with a representative result shown. All experiments in human cells were performed three times, with a representative result shown. Statistical analysis in Fig. 5a was performed using one way ANOVA and Tukey test for multiple comparisons.

### Cell lines
HeLa Flp-in TRex was a gift from Brian Raught, University of Toronto, Canada. Cells were cultured in Dulbecco's modified Eagle's medium (DMEM) supplemented with 10% fetal bovine serum (FBS) and 100 U/ml penicillin and streptomycin. Cells were maintained in a humidified, 5% $CO_2$ atmosphere at 37 °C. Cells were routinely tested for mycoplasma infection using MycoAlert (Lonza, Cat # LT07118).

We showed that Ubxn7 is more active than Faf1 in CMG unloading. This preference could not be explained by an intrinsic difference in ubiquitin binding. Instead, the UIM domain appears to be critical, since its deletion from Ubxn7 reduced this cofactor's activity and its addition to Faf1 enhanced Faf1 function. However, Faf1$^{UIM}$ was still less active than Ubxn7, and it supported much less

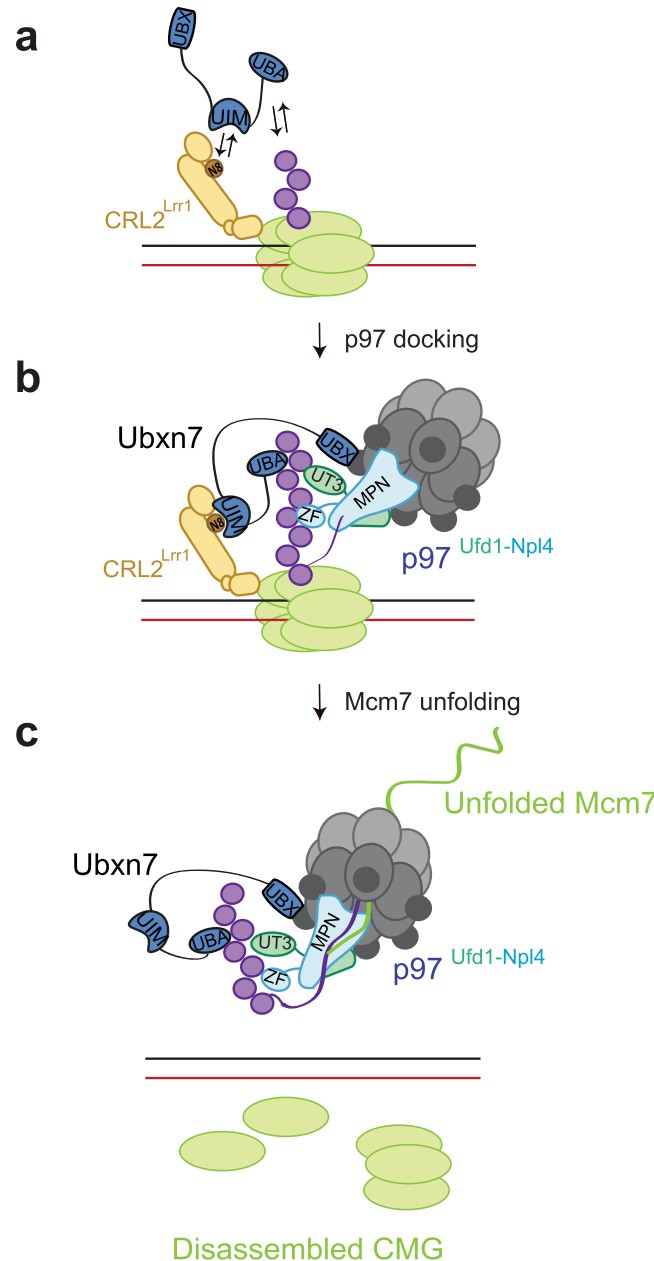

**Fig. 6 | A model for p97^Ufd1-Npl4-Ubxn7 assembly on ubiquitylated CMG. a** Ubxn7 binds weakly and transiently to ubiquitylated CMG-CRL2^Lrr1 prior to p97^Ufd1-Npl4 binding. **b** Multi-valent and cooperative interactions stabilize p97^Ufd1-Npl4-Ubxn7 binding to ubiquitylated CMG-CRL2^Lrr1, allowing initiation of ubiquitin chain unfolding. **c** Ubiquitylated Mcm7 is extracted from CMG and threaded through the central pore of p97.

## CRISPR/Cas9 knockout cell lines

The CRISPR/Cas9 gene-editing system was used to generate hUBXD7 knockout cell lines in HeLa Flp-in T-REX cells. The guide sequences for hUBXD7 (5′ to 3′ CACCGAATTAACCCCTTCAGCGCCG and AAACC GGCGCTGAAGGGGTTAATTC) (Supplementary Table 1) were cloned into the pX459 vector carrying hSpCas9 and transiently transfected into HeLa cells by using Lipofectamine 3000 (Thermo Fisher Scientific) according to the manufacturer's protocol. At 36 hrs post transfection, the cells were pulsed briefly with 1 μg/ml puromycin for a further 24 hrs. The surviving cells were then serially diluted to achieve 1 cell/well in 96-well plates for clonal selection. The knockout clones were verified by immunoblotting.

## DNA constructs

All DNA replication experiments were performed using 4.6 kb pKV45 plasmid[33] that contains 48x*lacO* repeats, with the exception of ChIP assay that was performed using pBluescript II (Agilent).

## *Xenopus* egg extracts and DNA replication

Animal work performed at Harvard Medical School was approved by the Harvard Medical Area Standing Committee on Animals (HMA IACUC Study ID IS00000051-6, approved 10/23/2020). The institution has

an approved Animal Welfare Assurance (D16-00270) from the NIH Office of Laboratory Animal Welfare. *Xenopus* egg extracts were prepared as described in the protocol[68]. For DNA replication, plasmids were licensed by incubation in high-speed supernatant of egg cytoplasm for 30 min at room temperature at a final concentration of 15 ng pKV45/μl extract (plasmid-pulldown assays) or 7.5 ng pBluescript II/ μl extract (chromatin immunoprecipitation assay). Replication was initiated by addition of two volumes of nucleoplasmic egg extract to licensing mixture. In all figures, the addition of nucleoplasmic egg extracts corresponds to the zero-minute time point. Where indicated, nucleoplasmic egg extracts were supplemented with p97i, NMS-873 (Sigma) or Culi, MLN4924 (Active Biochem), to a final concentration of 200 μM each and incubated for 5 min prior to initiation of DNA replication, unless otherwise indicated. Reactions were supplemented with approximately 140 nM recombinant FLAG-Ubxn7, 50–140 nM of recombinant FLAG-Faf1, and 80 nM recombinant Ufd1-Npl4.

## SDS-PAGE analysis and Western blotting

Samples were diluted with SDS Sample Buffer for a final concentration of 50 mM Tris [pH 6.8], 2% SDS, 0.1% Bromophenol blue, 10% glycerol, and 5% β-mercaptoethanol and resolved on Mini-PROTEAN precast gels (Bio-Rad) or home-made 6% polyacrylamide gels (Mcm7, Cdc45). Proteins were visualized with Instant Blue (Expedeon) stain. For Western blotting, gels were transferred to PVDF membranes (Perkin Elmer). Membranes were blocked in 5% nonfat milk in 1x PBST for 60 min at room temperature, then incubated with primary antibodies diluted to 1:300–1:20,000 in 1x PBST containing 1% BSA overnight at 4 °C. Membranes were then extensively washed with 1x PBST and incubated for 1 h at room temperature with goat anti-rabbit horseradish peroxidase-conjugated antibodies (Jackson ImmunoResearch, 111-035-003) at 1:10,000–1:30,000 dilution, light chain specific mouse anti-rabbit antibodies (Jackson ImmunoResearch, 211-032-171) at 1:10,000 dilution (all immunodepletion Western blots), rabbit anti-mouse horseradish peroxidase-conjugated antibodies (Jackson ImmunoResearch, 315-035-003) at 1:2,000 dilution, goat anti-rabbit at 1:10,000 (Promega, W4011) (for Fig. 5b and Supplementary Fig. 7f), or goat anti-mouse at 1:10,000 (Promega, W4021) (for Fig. 5b and Supplementary Fig. 7f), diluted in 5% nonfat milk in 1x PBST. Membranes were washed in 1x PBST, developed with ProSignal Pico ECL Spray (Genesee) or SuperSignal West Femto maximum sensitivity substrate (ThermoFisher) (for Mcm7, Sld5 (in all figures) and for Npl4, p97 (Fig. 4e) visualization), and imaged using an Amersham Imager 600 (GE Healthcare) or Bio-Rad Chemi-Doc Imaging system (BioRad). Where indicated, the contrast of the images was adjusted using Adobe Photoshop 22.5.7 Release or ImageJ 1.52q.

## Antibodies and immunodepletions

Rabbit polyclonal antibodies against the following proteins were used for Western blotting: FLAG (1:5,000)[30]; Mcm6 (1:5,000), Sld5 (1:5,000), Lrr1 (1:5,000), and Cul2 (1:5,000)[36]; Mcm7 (1:12,000)[69]; Cdc45 (1:20,000)[70]; Faf1 (1:5,000; Abcam, 202298), H3 (1:300; Cell Signaling, 9715 S), Ubxn7 (1:5,000; this study); hCDC45L (1:1,000, Novus Biologicals, NBP2-67897, Clone JJ091-04); hMCM3 (1:1,000, Cell Signaling Technology, 4012 S); hMCM7 (1:1,000, Cell Signaling Technology, 4018 S); hUBXD7 (1:1,000, Thermo Scientific, PA5-61972), Cyclin A

(H-432) (1:500, Santa Cruz, SC-751), Cyclin E (C-19) (1:500, Santa Cruz, SC-198), H3 (1:5,000, EMD Millipore, 06-755). Rabbit polyclonal antibodies against the following proteins were used for Flow Cytometry: hMCM7 (1:200, Bethyl Laboratories, A302-584A-M). Mouse monoclonal antibodies against the following proteins were used for Western blotting: GST (1:1,000; Cell Signaling, 2624, clone 26H1), ubiquitin (1:300; Santa Cruz Biotechnology, sc-8017, clone P4D1); ubiquitin (1:1,000, EMD Millipore, 04-263, clone FK2 (used in Fig. 6b), GAPDH (1:1,000, Santa Cruz Biotechnology, sc-47724, clone 0411). Rabbit polyclonal antibodies against p97 (1:5,000), Ufd1 (1:5,000), and Npl4 (1:5,000) used for Western blotting are a generous gift of Olaf Stemmann (University of Bayreuth, Germany) and were described previously[43]. Rabbit polyclonal antibodies raised against full-length *X. laevis* FLAG-Ubxn7, FLAG-Faf1, and Npl4 that were used for immunodepletions were prepared by Pocono Rabbit Farm and Laboratory. Ubxn7, Faf1, and Npl4 antibodies were then affinity-purified using AminoLink Coupling Resin (Thermo Scientific) with immobilized corresponding proteins prepared according to the manufacturer's protocol. For all depletions, 5 volumes of 1 mg/ml of corresponding affinity-purified antibodies were gently rotated with 1 volume of the protein A Sepharose Fast Flow beads (GE Healthcare) overnight at 4 °C. Five volumes of *Xenopus* egg extracts were then immunodepleted by three rounds of gentle rotation with 1 volume of antibody-bound beads at 4 °C. Egg extracts were then immunodepleted as described above.

### Plasmid pull-down assay
Plasmid pull-downs were performed as described previously[71]. Briefly, 4 μl of streptavidin-coated magnetic beads (Dynabeads M-280, Invitrogen) were incubated with 8 pmol of biotinylated LacR in 6 volumes of LacR-binding buffer (50 mM Tris-HCl [pH 7.5], 150 mM NaCl, 1 mM EDTA [pH 8.0], 0.02% Tween-20) for 40 min at room temperature. The beads were then extensively washed with 20 mM HEPES-KOH [pH 7.7], 100 mM KCl, 5 mM MgCl₂, 250 mM sucrose, 0.25 mg/ml BSA, and 0.02% Tween-20 and resuspended in 40 μl of the same buffer. 6 μl aliquots of replication reactions were mixed with pre-chilled beads at indicated time points and gently rotated for 30 min at 4 °C. The beads were then washed three times with 200 μl of wash buffer (20 mM HEPES-KOH [pH 7.7], 100 mM KCl, 5 mM MgCl₂, 0.25 mg/ml BSA, and 0.03% Tween-20), resuspended in 20 μl of 1xSDS Sample buffer, and boiled at 95 °C for 2 min prior to Western blotting.

### ChIP assay
ChIP were performed as described previously[72,73] with modifications. Briefly, at indicated time points, 2 μl of replication reactions were mixed with 48 μl of 1% formaldehyde in 20 mM HEPES-KOH [pH 7.7], 100 mM KCl, 5 mM MgCl₂, 250 mM sucrose and incubated for 5 min at room temperature. At exactly 5 min, crosslinking reaction was stopped by adding glycine to a final concentration of 125 mM. Samples were applied to Micro Bio-Spin™ P-6 Gel Columns (Bio-Rad) and spun at 1000 × g for 2 min. Eluate was collected and diluted 2-fold with 2x Sonication buffer containing 40 mM Tris-HCl pH 8.0, 200 mM KCl, 4 mM EDTA, 1 mM EGTA, 2% Triton X-100, 5 μg/mL Aprotinin (Roche), 5 μg/mL Leupeptin (Roche), 1 mM PMSF. Samples were then sonicated using the Qsonica Q800R3 at 4 °C, followed by 5-fold dilution with ice-cold ChIP buffer (20 mM Tris-HCl pH8.0, 2 mM EDTA, 187.5 mM NaCl, 1% Triton X-100, 12.5% glycerol).

Inputs and ChIP samples were then processed as described[74] with modifications. Specifically, for inputs, 50 μl of sonicated samples was diluted with 90 μl of ChIP buffer, supplemented with 0.5 μL of 4 mg/mL RNase A (Sigma), and incubated at 37 °C for 30 min. DNA was then ethanol precipitated, air-dried, and resuspended with 50 μl of 10% Chelex-100 (Bio-Rad) in MilliQ·H₂O by vortexing for 10 s. DNA samples were de-crosslinked by boiling for 10 min at 95 °C. Cooled samples were then supplemented with 0.5 μl of 20 mg/ml Proteinase K solution, and incubated at 55 °C, for 30 min in the thermomixer at 2000 rpm,

followed by boiling for 10 min at 95 °C. The beads were then pelleted by centrifugation at 12,000 × g for 1 min, and 30 μl of supernatant was transferred to a clean tube. The remaining beads were resuspended in 70 μl of MilliQ·H₂O, vortexed for 10 s, and centrifuged at 12,000 × g for 1 min. After centrifugation, 70 μl of supernatant was collected and pooled with the first supernatant and vortexed. 2 μl of 100%, 10%, and 1% of supernatant dilutions were used for quantitative PCR using PowerUp™ SYBR™ Green Master Mix (Applied Biosystems) to determine the standard curve using primers ChIP-F and ChIP-R (Supplementary Table 1).

For ChIP, 50 μl of sonicated samples was mixed with 2 μg of respective antibodies and rotated at 4 °C overnight. Next day, ChIP samples were supplemented with 10 μl of 30 mg/mL Dynabeads™ Protein A (Invitrogen) equilibrated into ChIP buffer and rotated for 2 h at 4 °C. The beads were next captured on the tube wall by placing the tubes in the pre-chilled magnetic racks, and the supernatant was completely removed. The beads were then washed sequentially with 500 μl of Low Salt Wash Buffer (20 mM Trish-HCl pH 8.0, 2 mM EDTA, 150 mM NaCl, 1% Triton X-100, 0.1% SDS), 500 μl of High Salt Wash Buffer (20 mM Trish-HCl pH 8.0, 2 mM EDTA, 500 mM NaCl, 1% Triton X-100, 0.1% SDS), 500 μl of LiCl Wash buffer (20 mM Trish-HCl pH 8.0, 2 mM EDTA, 250 mM LiCl, 1% NP-400, 0.1% sodium deoxycholate), and twice with 500 μl TE (10 mM Tris-HCl pH 8.0, 1 mM EDTA). The beads were then resuspended in 100 μl of the TE buffer and transferred to a new tube, followed by complete removal of the TE buffer. The beads were then resuspended with 50 μl of 10% Chelex-100 (Bio-Rad) in MilliQ·H₂O. DNA form ChIP samples was then de-crosslinked, purified, and eluted as described for the input samples above. 2 μl of purified DNA samples was then used for quantitative PCR using primer pair (ChIP-F and ChIP-R) to determine percent of the input. The experiment shown in Supplementary Fig. 2b was repeated four times for time points 0 and 5 min, and three times for time points 15 and 20 min.

### Expression and purification of recombinant *Xenopus* Ubxn7
The pGEX-6p-1_GST-TEV-FLAG-Ubxn7 plasmid used for expression of *Xenopus* Ubxn7 (a generous gift of Emily Low) was constructed as follows. The *X. laevis* ubxn7 ORF was amplified from *Xenopus* cDNA using primers ELO32_Ubxn7_fwd and ELO33_Ubxn7_rev (Supplementary Table 1) and FLAG tag with TEV protease cleavage site were inserted at 3′ end of the *ubxn7* sequence. The amplified *ubxn7* product and pGEX-6p-1 backbone were assembled in one construct using NEBuilder HiFi DNA Assembly Master Mix (New England Biolabs). To eliminate expression of un-tagged Ubxn7, ATG codon of the *ubxn7* ORF was removed by "Round-the-Horn" mutagenesis using primers OK7 and OK8 (Supplementary Table 1). The resulted pGEX-6p-1_GST-TEV-FLAG-Ubxn7^ΔMet1 plasmid was used as a template to construct Ubxn7^ΔUBA, Ubxn7^UIM*, and Ubxn7^UBX* mutants by "Round-the-Horn" mutagenesis using primer pairs OK8/OK58, OK9/10, and OK11/12 (Supplementary Table 1), respectively. To generate Ubxn7^UBA*, first, I46A substitution was introduced by "Round-the-Horn" mutagenesis using primers OK18/OK19, and then L65A and L73A were generated using primers OK15/OK16 (Supplementary Table 1). All mutations and truncation were confirmed by Sanger sequencing.

GST-TEV-FLAG-Ubxn7 was expressed in T7 Express cells (New England Biolabs) by induction with 0.5 mM IPTG for 4 hrs at 37 °C in LB growth media. Bacterial pellets were sonicated in 50 mM Tris-HCl [pH 7.5], 250 mM NaCl, 0.1 mM EDTA, 0.1 mM EGTA, 1 mM DTT, 1 mM PMSF, 0.4% Triton X-100, 1x EDTA-free cOmplete protease inhibitor cocktail (Roche), following centrifugation at 21,000 × g for 1 h at 4 °C. The soluble fraction was collected and incubated with the Glutathione Sepharose 4B resin (GE Healthcare) for 1 hr at room temperature with end-over-end rotation. The resin was washed three times with Wash buffer (50 mM Tris-HCl [pH 7.5], 250 mM NaCl, 0.1 mM EDTA, 0.1 mM EGTA, 1 mM DTT). GST-TEV-FLAG-Ubxn7 was then eluted with Elution buffer (50 mM Tris-HCl [pH 8.0], 150 mM NaCl, 10 mM reduced

glutathione, 1 mM DTT), and Ubxn7-containing elutions were combined and concentrated on Amicon Ultra-15 10 K concentrator (Millipore). To remove the GST tag, TEV protease was added to the concentrated elutions, and the mixture was incubated overnight at 4 °C with end-over-end rotation. GST, TEV protease, and FLAG-Ubxn7 were then separated on a HiLoad 16/600 Superdex 75 pg column (GE Healthcare) and simultaneously exchanged into Storage buffer (50 mM Tris-HCl [pH 7.5], 150 mM NaCl, 1 mM DTT, 10% glycerol). The fractions corresponding to the Ubxn7 peak were pooled, concentrated on Amicon Ultra-15 10 K concentrator (Millipore), and flash frozen in liquid nitrogen, then stored at −80 °C.

### Expression of FLAG-Ubxn7 in wheat germ extract cell-free protein expression system

The pF3A-WG(BYDV)-Flexi-FLAG-Ubxn7$^{WT}$, pF3A-WG(BYDV)-Flexi-FLAG-Ubxn7$^{ΔUBA}$, and pF3A-WG(BYDV)-Flexi-FLAG-Ubxn7$^{ΔUBA/UIM*}$ plasmids used for expression of *Xenopus* Ubxn7 in wheat germ extracts were constructed as follows. DNA sequences encoding FLAG-Ubxn7$^{ΔMet1}$ or FLAG-Ubxn7$^{ΔUBA}$ were amplified using primer pair OK83/ OK84 (Supplementary Table 1) and pGEX-6p-1_GST-TEV-FLAG-Ubxn7$^{WT}$ or pGEX-6p-1_GST-TEV-FLAG-Ubxn7$^{ΔUBA}$ as templates, respectively. The amplified products and the pF3A-WG (BYDV)-Flexi (Promega) backbone digested with AsiSI and EcoRI (New England Biolabs) were assembled in one construct using NEBuilder HiFi DNA Assembly Master Mix (New England Biolabs). The insertions were confirmed by diagnostic restriction digest and Sanger sequencing. The Ubxn7$^{ΔUBA/UIM*}$ mutant was then constructed by "Round-the-Horn" mutagenesis using primer pair OK9/10 (Supplementary Table 1) and pF3A-WG(BYDV)-Flexi-FLAG-Ubxn7$^{ΔUBA}$ as a template. All mutations and truncations were confirmed by Sanger sequencing.

For production of Ubxn7 in wheat germ extract, 300 ng of pF3A-WG(BYDV)-Flexi-FLAG-Ubxn7$^{WT}$, pF3A-WG(BYDV)-Flexi-FLAG-Ubxn7$^{ΔUBA}$, or empty vector were incubated with 3 µl of TnT® SP6 High-Yield Protein Expression System (Promega) in a total of volume of 5 µl for 2 h at 25 °C.

### Expression and purification of recombinant *Xenopus* Faf1 and Faf1$^{UIM}$

The *X. laevis faf1* ORF was amplified from Xenopus cDNA using primers OK21/OK22 (Supplementary Table 1). The amplified *faf1* product and pGEX-6p-1_GST-TEV-FLAG backbone derived from pGEX-6p-1_GST-TEV-FLAG-Ubxn7 were assembled in one construct using NEBuilder HiFi DNA Assembly Master Mix (New England Biolabs). To generate the FLAG-Faf1$^{UIM}$ chimera, the pGEX-6p-1_GST-TEV-FLAG-Faf1 backbone was first amplified using primers OK85/OK86 (Supplementary Table 1) to delete the DNA encoding amino acids 241−368 of the *Xenopus* Faf1 sequence. The resulting PCR product was then assembled with the synthesized chimeric sequence containing DNA residues 919−984 of the *Xenopus* Ubxn7 cDNA inserted between DNA residues 879−882 of the *Xenopus* Faf1 cDNA using NEBuilder HiFi DNA Assembly Master Mix (New England Biolabs). GST-TEV-FLAG-Faf1 and GST-TEV-FLAG-Faf1$^{UIM}$ were expressed in T7 Express cells (New England Biolabs) by induction with 0.5 mM IPTG overnight at 16 °C in LB media. FLAG-Faf1 and FLAG-Faf1$^{UIM}$ were then purified according to the FLAG-Ubxn7 purification protocol described above.

### Expression and purification of recombinant *Xenopus* Ufd1-Npl4

Construct for co-expression of GST-3C-Ufd1 and Npl4 was prepared by assembly of the synthesized and codon-optimized *X. laevis ufd1* and *nploc4* ORFs (interspaced by the T7 promoter, lac operator, and ribosome binding site sequences) into SalI- and HindIII-digested pOPINK backbone using NEBuilder HiFi DNA Assembly Master Mix (New England Biolabs). To generate pOPINK-GST-3C-Ufd1$^{ΔUT3}$_Npl4$^{LV}$ construct, pOPINK-GST-3C-Ufd1_Npl4 was digested with BseRI and HindIII, gel-purified, and the cut-out 797-nt fragment was replaced with the synthesized sequence containing T592L/F593V substitutions using NEBuilder HiFi DNA Assembly Master Mix (New England Biolabs). To

delete 2-642 nt of the *ufd1* sequence, the resulted plasmid was amplified with OK76/OK77 primers, gel-purified, and ligated using T4 DNA ligase (New England Biolabs). To introduce L240A/W243A/R244E substitutions, pOPINK-GST-3C-Ufd1_Npl4 was first amplified with OK59/OK60 primers to delete the MPN domain, the resulted PCR product was then assembled with the synthesized mutant MPN domain sequence using NEBuilder HiFi DNA Assembly Master Mix (New England Biolabs).

GST-3C-Ufd1 and Npl4 proteins were co-expressed in Rosetta 2 (DE3) (Novagen) by induction with 0.5 mM IPTG overnight at 16 °C in LB media containing 10 µM ZnSO$_4$. Bacterial pellets were sonicated in UN-Lysis buffer (25 mM HEPES-KOH [pH 8.0], 300 mM KCl, 1 mM MgCl$_2$, 10 µM ZnSO$_4$, 1 mM DTT, 1 mM PMSF, 0.4% Triton X-100, 1x EDTA-free cOmplete protease inhibitor cocktail (Roche)), following centrifugation at 21,000 × g for 1 h at 4 °C. The soluble fraction was collected and incubated with the Glutathione Sepharose 4B resin (GE Healthcare) for 1 h at room temperature with end-over-end rotation. The resin was washed once with UN-Wash buffer-ATP (25 mM HEPES-KOH [pH 8.0], 300 mM KCl, 1 mM MgCl$_2$, 10 µM ZnSO$_4$, 2 mM ATP, 1 mM DTT, 10% sucrose), following two washes with UN-Wash buffer (25 mM HEPES-KOH [pH 8.0], 300 mM KCl, 1 mM MgCl$_2$, 10 µM ZnSO$_4$, 1 mM DTT, 10% sucrose). Beads were then resuspended with one volume of UN-Wash buffer, supplemented with GST-tagged PreScission Protease (a generous gift of Tycho Mevissen), and were incubated overnight at 4 °C with end-over-end rotation. Eluate containing the Ufd1-Npl4 heterodimer was collected, and 3 µl aliquots were flash frozen in liquid nitrogen, then stored at −80 °C.

GST-tagged variants of Ufd1-Npl4 were purified as above, but instead of GST tag cleavage with PreScission Protease, proteins were eluted from the Glutathione Sepharose 4B resin with 25 mM HEPES-KOH [pH 8.0], 300 mM KCl, 1 mM MgCl$_2$, 10 µM ZnSO$_4$, 1 mM DTT, 10 mM reduced glutathione, 10% sucrose buffer. Good elutions were combined, dialyzed overnight into UN-storage buffer (25 mM HEPES-KOH [pH 8.0], 300 mM KCl, 1 mM MgCl$_2$, 10 µM ZnSO$_4$, 1 mM DTT, 10% sucrose), concentrated on Amicon Ultra-15 3 K concentrator (Millipore), and flash frozen in liquid nitrogen, then stored at −80 °C.

A GST-3C-Npl4 expressing construct and Npl4 antigen for raising anti Npl4 antibodies were prepared as follows. pOPINK-GST-3C-Ufd1_Npl4 was amplified using primers OK54/OK55 such that the *ufd1* sequence was deleted, followed by blunt end ligation. GST-3C-Npl4 was then expressed in Rosetta 2 (DE3) (Novagen) by induction with 0.5 mM IPTG overnight at 16 °C in LB media containing 10 µM ZnSO$_4$. Bacterial pellets were sonicated in UN-Lysis buffer, following centrifugation at 21,000 × g for 1 h at 4 °C. The soluble fraction was collected and incubated with the Glutathione Sepharose 4B resin (GE Healthcare) for 1 h at room temperature with end-over-end rotation. The resin was washed three times with UN-Wash buffer, and protein was eluted by incubation with one resin bed volume of UN-Wash buffer + 10 mM reduced glutathione, the procedure was repeated six times. Elutions containing GST-3C-Npl4 were combined and supplemented with PreScission Protease (a generous gift of Tycho Mevissen), and incubated overnight at 4 °C with end-over-end rotation. Proteins were then resolved by a horizontal SDS-PAGE and stained with InstantBlue (Expedeon). The band corresponding to Npl4 was excised, and the protein was electroeluted in 1x SDS-PAGE buffer. Protein was concentrated on Amicon Ultra-15 10 K concentrator (Millipore) and flash frozen in liquid nitrogen, then stored at −80 °C.

### Expression and purification of His-p97

The plasmid used for expression of *Xenopus* His-p97 was a generous gift of Olaf Stemmann (University of Bayreuth, Germany) and was described previously[43]. His-p97 was expressed in Rosetta 2 (DE3) (Novagen) by induction with 0.5 mM IPTG overnight at 16 °C in LB media. The bacterial pellet was resuspended in p97-Lysis buffer (25 mM HEPES-KOH [pH 8.0], 1 M NaCl, 5 mM MgCl$_2$, 10 mM imidazole,

5 mM β-mercaptoethanol, 1 mM PMSF, 0.4% Triton X-100, 1 mg/ml lysozyme (Sigma), 0.06 U/μL benzonase (Sigma), 1x EDTA-free cOmplete protease inhibitor cocktail (Roche)), followed by incubation on ice for 20 min. Cells were then sonicated and insoluble material was pelleted by centrifugation at 21,000 × $g$ for 1 h at 4 °C. The soluble fraction was filtered by passage through a 0.45 μm syringe filter, degassed and applied to an equilibrated 5 ml HisTrap HP column (Cytiva). The column was washed with 5-column volumes of p97-Lysis buffer and 10-column volumes of salt reduction buffer (25 mM HEPES-KOH [pH 8.0], 300 mM NaCl, 5 mM MgCl$_2$, 20 mM imidazole, 5 mM β-mercaptoethanol). His-p97 was then eluted with a 20-column volume linear gradient of 20–350 mM imidazole in salt reduction buffer. Peak fractions were collected, concentrated, and a hexameric form of His-p97 was then separated on a Superose 6 Increase 10/300 GL column (GE Healthcare) and simultaneously exchanged into Storage buffer (25 mM HEPES-KOH [pH 8.0], 100 mM NaCl, 5 mM MgCl$_2$, 1 mM DTT, 10% sucrose). The fractions corresponding to the His-p97 peak were pooled, concentrated on Amicon Ultra-15 100 K concentrator (Millipore), and flash frozen in liquid nitrogen, then stored at −80 °C.

### Synthesis of K48-linked polyubiquitin chains in vitro

To generate K48-linked polyubiquitin chains for the experiment shown in Fig. 4c, human ubiquitin (R&D Systems) at a final concentration of 1.7 mM was incubated overnight at 37 °C with 0.333 μM UBE1 (R&D Systems) and 3.33 μM Ube2R1 (R&D Systems) in reaction buffer containing 50 mM Trish-HCl [pH 8.0], 10 mM ATP, 10 mM MgCl$_2$, 1 mM DTT. Ubiquitylation reaction was then diluted ~28-fold with Buffer A (50 mM NH$_4$Ac pH 4.5), and loaded on a MonoS 5/50GL column (GE Healthcare). Ubiquitin chains were eluted with a linear gradient (5–100%) of Buffer B (50 mM NH$_4$Ac pH4.5, 1 M NaCl), and peak fractions were combined, concentrated on Amicon Ultra-15 3 K concentrator (Millipore) and simultaneously exchanged into storage buffer (10 mM Tris-HCl pH 7.6), then flash frozen in liquid nitrogen and stored at −80 °C.

### Immunoprecipitations of FLAG-Ubxn7 and FLAG-Faf1

For Supplementary Fig. 1f, 32 μl of 2 μM FLAG-Ubxn7 variants were incubated with 5 μl of anti-FLAG resin (Sigma) for 2 h at 4 °C. The beads were washed 4 times with 500 μl of medium stringency IP buffer (20 mM HEPES [pH 7.7], 100 mM KCl, 5 mM MgCl$_2$, 0.1% NP-40). 40 μl of 5.8 μM K48-diubiquitin chains (R&D Systems) diluted in medium stringency IP buffer were added to the beads and rotated on a wheel at room temperature for 30 min. The beads were washed 4 times with 500 μl of medium stringency IP buffer. Samples were eluted by incubating the resin with 50 μL of IP buffer + 1 mg/mL 3xFLAG peptide for 1 h at room temperature, mixed with 1 volume of 2x SDS sample buffer, and analyzed by Western blotting as described above.

For Supplementary Fig. 1g, FLAG-Ubxn7 (30 nM final concentration) was incubated in 17.8 μL of 35% nucleoplasmic extract diluted in IP buffer (10 mM HEPES [pH 7.7], 50 mM KCl, 2.5 mM MgCl$_2$, 0.1% NP-40, 0.1 mg/mL BSA) or IP buffer supplemented with Culi (200 μM final concentration) or p97i (200 μM final concentration) for 30 min at room temperature. 16 μL of each reaction was added to 5 μL of anti-FLAG resin (Sigma) and rotated on a wheel at room temperature for 30 min. The beads were washed 5 times with 100 volumes of IP buffer. Samples were eluted by incubating the resin with 20 μl of IP buffer + 1 mg/mL 3xFLAG peptide for 1 hr at room temperature and mixed with 1 volume of 2x SDS sample buffer. Samples were analyzed by SDS-PAGE and Western blotting as described above. Experiment in Fig. 3b was performed as described above, but the beads were washed either with low (10 mM HEPES [pH 7.7], 50 mM KCl, 2.5 mM MgCl$_2$, 0.1% NP-40, 0.1 mg/mL BSA) or medium (20 mM HEPES [pH 7.7], 100 mM KCl, 5 mM MgCl$_2$, 0.1% NP-40, 0.1 mg/mL BSA) stringency IP buffer. The experiment in Supplementary Fig. 6b was performed as described

above, but FLAG-Ubxn7 and FLAG-Faf1 variants (30 nM final concentration) were incubated in 17.8 μL nucleoplasmic extract supplemented with Culi/p97i mix (200 μM final concentration each) or p97i (200 μM final concentration), and the beads were washed with medium stringency IP buffer.

For Fig. 4c, 32 μl of FLAG-Ubxn7 or FLAG-Faf1 (0.16 μg/μl final concentration) were incubated with 6 μl of anti-FLAG resin (Sigma) for 1 h at 4 °C. The beads were washed 4 times with 500 μl of medium stringency IP buffer (20 mM HEPES [pH 7.7], 100 mM KCl, 5 mM MgCl$_2$, 0.1% NP-40). 80 μl of K48-linked ubiquitin chains (final concentration 0.6 μg/μl) diluted in medium stringency IP buffer were added to the beads and rotated on a wheel at room temperature for 30 min. The beads were washed 4 times with 500 μl of medium stringency IP buffer. Samples were eluted by incubating the resin with 50 μL of IP buffer + 1 mg/mL 3xFLAG peptide for 1 h at room temperature, mixed with 1 volume of 2x SDS sample buffer, and analyzed by Western blotting as described above.

For Supplementary Fig. 5c, 20 μl of FLAG-Ubxn7 or FLAG-Faf1 (0.24 μg/μl final concentration) were incubated with 5 μl of anti-FLAG resin (Sigma) for 1 h at 4 °C. The beads were washed 4 times with 500 μl of low stringency IP buffer (10 mM HEPES [pH 7.7], 50 mM KCl, 2.5 mM MgCl$_2$, 0.1% NP-40). Then, 20 μl of different tri- or tetraubiquitin chains (final concentration 0.345 μg/μl) were added to the beads and rotated on a wheel for 60 min at room temperature. The beads were washed 4 times with 500 μl of low stringency IP buffer. Samples were eluted by incubating the resin with 20 μL of IP buffer + 1 mg/mL 3xFLAG peptide for 1 h at room temperature, mixed with 1 volume of 2x SDS sample buffer, and 12 μl of each eluate was resolved on a polyacrylamide gel and visualized by staining with InstantBlue (Expedeon).

### GST pull-down assay

For Supplementary Fig. 2d, 60 μL of GST-Ufd1 and Npl4 variants or GST (Genescript) (0.05 μg/μL final concentration in UN-storage buffer) were incubated with 20 μL of the Glutathione Sepharose 4B resin (GE Healthcare) on a wheel for 1 h at 4 °C. The beads were then washed 4 times with 50 volumes of IP buffer (10 mM HEPES [pH 7.7], 50 mM KCl, 2.5 mM MgCl$_2$, 0.1% NP-40, 0.1 mg/mL BSA), resuspended in 110 μL of wash buffer and split between 4 tubes (30 μl, each). Next, 20 μl of IP buffer or indicated extracts were added to the beads. The reactions were then rotated on a wheel at room temperature for 1 h, washed 4 times with 200 μL of IP buffer, resuspended in 100 μL 1x SDS sample buffer, and boiled for 2 min at 95 °C. Samples were analyzed by SDS-PAGE and Western blotting as described above. For Supplementary Fig. 3b, Ubxn7/Npl4-depleted extracts were first supplemented with 1 volume of wheat germ extract expressing Ubxn7 variants or containing an empty vector. For Fig. 3c, Npl4-depleted extracts were first pre-incubated for 30 min at room temperature with matching buffer, p97i, mixture of p97i and Culi, or a combination of p97i, E1i (TAK-243) (selleckchem) (200 μM final concentration), and catalytic domain of Usp2 (R&D Systems) (7 μM final concentration), as indicated. To inhibit Usp2 activity, egg extracts were supplemented with a deubiquitinase inhibitor, Ub-VS (R&D Systems) (25 μM, final concentration), or matching buffer (50 mM MES, pH 6.0) and incubated for 30 min at room temperature. Following that, extracts were supplemented with a mixture of recombinant K48-linked ubiquitin chains to a final concentration of 0.5 μg/μl, as indicated.

### Reconstitution of the p97$^{Ufd1-Npl4-Ubxn7}$ complex with purified proteins in vitro

For Supplementary Fig. 4b, 10 nM GST-Ufd1/Npl4 or GST was pre-bound to 4 μl of the Glutathione Sepharose 4B resin (GE Healthcare) in a total volume of 60 μl by rotating on a wheel for 1 h at 4 °C. The beads were then washed 4 times with 20 mM HEPES [pH 7.7], 100 mM KCl, 5 mM MgCl$_2$, 0.1% NP-40, 0.1 mg/mL BSA. The void volume was removed and beads were mixed with 50 μl of 10 nM His-p97 (as a

hexamer), 10 nM FLAG-rUbxn7, 0.8 μM K48-linked ubiquitin chains (2-7) (R&D Systems) in reaction buffer containing 25 mM HEPES, [pH 7.6], 100 mM NaCl, 50 mM MgCl₂, 1 mM DTT, 2 mM ATP, 10 μM NMS-873 (Sigma). Reactions were then rotated on a wheel for 1 h at room temperature, washed 4 times with 20 mM HEPES [pH 7.7], 100 mM KCl, 5 mM MgCl₂, 0.1% NP-40, 0.1 mg/mL BSA, resuspended in 100 μl of 1x SDS sample buffer, and boiled for 2 min at 95 °C. Samples were analyzed by SDS-PAGE and Western blotting as described above.

For Supplementary Fig. 4c, 10 nM FLAG-Ubxn7, 10 nM GST-Ufd1/Npl4, 10 nM His-p97 (as a hexamer), 0.8 μM K48-linked ubiquitin chains (2-7) (R&D Systems), 25 mM HEPES, [pH 7.6], 100 mM NaCl, 50 mM MgCl₂, 1 mM DTT, 2 mM ATP, 10 μM NMS-873 (Sigma) were incubated for 1 h at room temperature in a total volume of 50 μl. Reconstitution reaction was then mixed with 5 μl of anti-FLAG resin (Sigma) and incubated on a wheel at 4 C for 1 h. The beads were washed 5 times with 20 mM HEPES [pH 7.7], 100 mM KCl, 5 mM MgCl₂, 0.1% NP-40, 0.1 mg/mL BSA. Samples were eluted by incubating the resin with 30 μL of IP buffer + 1 mg/mL 3xFLAG peptide for 1 h at room temperature, and elutions were mixed with the equal volume of 2× SDS sample buffer, and boiled for 2 min at 95 °C. Samples were analyzed by SDS-PAGE and Western blotting as described above.

### MCM7 loading on chromatin by flow cytometry

For flow cytometry, $3.5 \times 10^5$ cells/well were seeded in a 6-well plate. p97 inhibition was carried out using CB-5083 (Cayman Chemicals), which was added to cells at a final concentration of 10 μM for 8 h before harvesting. Chromatin extraction and fixation of HeLa Flp-in T-REX cells for flow cytometry were performed as described previously[52]. Briefly, cells were trypsinized and resuspended into DMEM containing 10% FBS as single cells. Cells were centrifuged at $2000 \times g$ for 5 min and resuspended in PBS and centrifuged again at $2000 \times g$ for 3 min. Cells were lysed in 500 ml of ice-cold CSK buffer (300 mM Sucrose, 300 mM NaCl, 3 mM MgCl₂, 1 mM PIPES pH 7.0) containing 0.5% Triton X-100, HALT protease inhibitors (Thermo Scientific) and phosphatase inhibitor (1 mM sodium orthovanadate) for 7 min on ice. 1 ml of 1% BSA-PBS was added to the samples, mixed, and cells were centrifuged at $2000 \times g$ for 3 min. Cells were fixed by resuspending thoroughly in 4% paraformaldehyde (Electron Microscopy Sciences) in PBS for 15 min at room temperature in the dark. 1 ml of 1% BSA-PBS was added, mixed by gentle pipetting and cells were collected at $2000 \times g$ for 7 min. Samples were resuspended in primary antibody (1:200 μl for MCM7) diluted in 1% BSA-PBS containing 0.1% NP-40 and rocked overnight at 4 °C in the dark. 1 ml of 1% BSA-PBS containing 0.1% NP-40 was added, cells were mixed gently and collected at $2000 \times g$ for 7 min. Cells were resuspended in 300 μl of secondary goat anti-rabbit antibody (Life Technology, Alexa Fluor-488, A-11008, 1:1,000) diluted in 1% BSA-PBS containing 0.1% NP-40. Cells were incubated by rocking at room temperature for 1 h in the dark. 1 ml of 1% BSA-PBS was added, cells were mixed gently, and cells were collected at $2000 \times g$ for 7 min. Finally, cells were resuspended in 1% BSA-PBS containing 0.1% NP-40 with 5 μg/ml of Propidium Iodide (PI) (Sigma) and 200 μg/ml RNAse A (Sigma) for 30 min at room temperature in the dark to label DNA.

For flow cytometric analysis, samples were analyzed on a LSR II flow cytometer (BD Biosciences) equipped with 561 and 488-nm lasers. Live cells were gated using forward and side scatter. Single cells were gated using PI area vs PI height. 3000 single cells were analyzed for each of the three independent experiments. MCM7 loading (488 nm) versus DNA content (PI, 561 nm) were plotted to identify levels of MCM loading on chromatin at the end of S phase and beginning of G2/M for each sample. Data was analyzed using Flowjo version 10.8 (Tree Star).

For flow cytometry analysis in Supplementary Figs. 7e, $3.5 \times 10^5$ cells/well were seeded in a 6-well plate. Briefly, cells were trypsinized and resuspended into PBS as single cells. Cells were centrifuged at $400 \times g$ for 5 min and resuspended in PBS. Cells were fixed by vortexing gently while adding ice-cold ethanol dropwise and incubated overnight at −20 °C. Cells were then washed three times with PBS and resuspended in Propidium Iodide (PI) (Sigma) and 200 μg/ml RNAse A (Sigma) for 30 min at room temperature in the dark to label DNA.

### Cell synchronization and chromatin fractionation

Cells were synchronized at the G1/S transition by double thymidine block. Cells were treated with 2.5 mM thymidine (Sigma) for 17 h, washed three times in DMEM containing 10% FBS and released into drug-free medium for 8 hrs and treated again with 2.5 mM thymidine for 16 h. Cells were washed three times with DMEM containing 10% FBS and released into medium containing 10 μM RO-3306 (Sigma)[53] CDK1 inhibitor for 10 h to capture cells at the end of S phase and prevent entry into mitosis. Cells were collected by trypsinization and fractionated into soluble and chromatin fractions as described previously[75]. Briefly, cells were lysed in 100 ml ice-cold Buffer A (10 mM HEPES pH 7.9, 10 mM KCl, 1.5 mM MgCl₂, 0.34 M sucrose, 10% glycerol, 1 mM DTT) containing HALT protease inhibitors and 0.1% Triton X-100. Lysates were rocked at 4 °C for 10–15 min and centrifuged at $1300 \times g$ for 10 min. The supernatant containing the soluble fraction was removed, and the pellet was washed gently in Buffer A without detergent, resuspended in 100 ml ice-cold Buffer B (3 mM EDTA, 0.2 mM EGTA, 1 mM DTT and protease inhibitors) and rocked at 4 °C for 10–15 min and centrifuged at $2000 \times g$ for 10 min. The pellet (chromatin) fraction was resuspended in Buffer A containing 1 mM CaCl₂ and briefly sonicated prior to the addition of 0.2 U Benzonase (Sigma). Samples were incubated for 30 min at 37 °C and then spun at $16000 \times g$ for 10 min. The supernatant was the chromatin-bound fraction. Total protein from soluble and chromatin fractions were estimated by Bradford assay (BioRad), and 25 μg total protein was resolved by SDS-PAGE and immunoblotted with the appropriate antibodies.

### Cyclin levels by immunoblot analysis

Cells were synchronized at the G1/S transition by double thymidine block as above, but released into medium without CDK1 inhibitor. Cells were collected by trypsinization and lysed in RIPA cell lysis buffer [25 mM Tris-Cl pH 7.1, 150 mM NaCl, 1% NP-40, 1% Sodium deoxycholate, 0.1% SDS, Halt protease inhibitor (Pierce)]. Cells were incubated at 4 °C for 10 min and then centrifuged at $19000 \times g$ for 15 min. The supernatant was removed, and the protein concentration was estimated using the DCA assay (Bio-Rad), and proteins were resolved by SDS-PAGE and immunoblotted with the appropriate antibodies.

### Reporting summary

Further information on research design is available in the Nature Research Reporting Summary linked to this article.

## Data availability

All materials used in this study are available commercially or will be made available upon request without any restrictions from the corresponding author. All data supporting the findings of this study are available within the article and its Supplementary files. Unprocessed and uncropped gel and Western blot images are provided as Source Data with this paper. Source data are provided with this paper.

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

## Acknowledgements

We thank Tom Rapoport, Wade Harper, Zhejian Ji, Ben Stinson, and Tycho Mevissen for comments on the manuscript. We are also grateful to Olaf Stemmann for kindly providing antibodies against *Xenopus* Ufd1 and Npl4 used for Western blotting, Karen Adelman for allowing access to Qsonica Q800R3 and the CFX384 Touch Real-Time PCR System, Yang Lim for designing and providing primers for ChIP and for initial ChIP protocol optimization, and Emily Low for constructing the pGEX-6p-1_GST-TEV-FLAG-Ubxn7 vector. J.C.W. is supported by NIH grant GM80676. M.R. is supported by NIH grant GM127557. J.C.W. is a Howard Hughes Medical Institute Investigator and an American Cancer Society Research Professor.

## Author contributions

O.V.K. performed all experiments with the exception of those performed in human cells, which were carried out by S.M. under the supervision of M.R. J.C.W. and O.V.K. conceived experiments, analyzed data, and wrote the paper with input from S.M. and M.R.

## Competing interests

J.C.W. is a co-founder of MoMa therapeutics, in which he has a financial interest. The remaining authors declare no competing interests.
