## [Peer Review File · Nature Communications]

Cooperative assembly of p97 complexes involved in replication terminationREVIEWER COMMENTS

Reviewer #1 (Remarks to the Author):

The study by Kochenova et al. investigates how the p97 ATPase disassembles the replicative CMG helicase upon its ubiquitination during DNA replication termination. Using *Xenopus* egg extracts, they found that the p97 co-factor Ubxn7 is required for efficient CMG unloading in a manner that depends on its ability to interact with ubiquitin chains, neddylated CRL complexes and p97 via the UBA, UIM and UBX domains, respectively. They show that recruitment of Ubxn7 and p97/Npl4-Ufd1 to CMG is interdependent, and this cooperative assembly involves interactions between Ubxn7, p97, Ufd1-Npl4 and ubiquitin, but not a p97 substrate per se, a mechanism that seems likely to apply more broadly to interactions between p97-co-factor complexes and their ubiquitinated targets. The authors also provide evidence to suggest that Faf1, another p97 UBA-UBX co-factor, contributes to CMG unloading in the absence of Ubxn7, and that Ubxn7 is also important for CMG disassembly in human cells.

The new findings showing cooperative assembly of p97 and its co-factors on ubiquitin chains involving multivalent interactions are potentially important, for the most part convincing and should be of broad interest. I therefore believe the manuscript is appropriate in principle for *Nature Communications*. However, there are several aspects of the study that need to be strengthened significantly before publication in this journal can be recommended. Specifically, the following points should be addressed:

- 1) The inability of Npl4 LV-Ufd1 deltaUT3 to support CMG disassembly in Npl4-depleted extracts (Figure 2) is not convincing. First, Extended Data Figure 2a,c indicate that much less recombinant Ufd1 deltaUT3 mutant was present relative to the WT protein. Second, the deltaUT3 mutation deletes more than half of the Ufd1 protein, which might impair Npl4-Ufd1 function in ways that are not solely due to loss of ubiquitin binding. The authors could attempt to generate and test a Ufd1 mutant that abrogates ubiquitin binding without removing a large portion of the protein.
- 2) Based on corresponding point mutations in yeast Npl4, the authors state that the xNpl4 AAE triple point mutant is expected to lose ubiquitin unfolding activity (Figure 2). More direct evidence for this claim should be provided.
- 3) It would be informative to probe for Faf1 in the experiments in Figure 1a and 2a,c.
- 4) The data in Figure 4 are somewhat underwhelming. The authors generated a Faf1 chimera containing an insertion of the Ubxn7 UIM motif to test the idea that Ubxn7 is more active for CMG unloading due to CRL2/Lrr1 interaction via the UIM. However, they did not provide evidence that this actually enables Faf1 to interact with CRL2/Lrr1. The authors should also test whether the UIM insertion impacts Faf1's ability to interact with p97, a possibility they discuss on page 16. The ability of the Faf1/UIM chimera to support p97/Npl4-Ufd1 recruitment in the presence of Ub-VS (Figure 4e) is so weak that it seems questionable this has any real significance. In general, the notion that Faf1/UIM is less active than Ubxn7 in unloading CMG through failure to restrain a deubiquitinating enzyme acting on Mcm7 does not seem well supported by the presented data.
- 5) Does Faf1's UBA and UBX domains promote cooperative assembly of p97/Ufd1-Npl4-Faf1 on ubiquitin chains, similar to Ubxn7?
- 6) The examination of Faf1's role in CMG unloading should be extended by analyzing the impact of FAF1 depletion in Wild-type and UBXD7 KO cells (Figure 5a,b). It would also be relevant to test the ability of WT UBXD7 and UBA and/or UBX mutants to rescue the CMG disassembly defect of UBXD7 KO cells via stable reconstitution, which should be well feasible given that UBXD7 was knocked out in a Flp-in background.

Minor points:

7) Does the UBX* mutation affect Ubxn7 interaction with K48-Ub(2) in vitro, as tested for the UBA* mutant in Extended Data Figure 1f?

8) The authors found that Ubxn7 binds Cul2 in a UIM-dependent manner (Extended Data Figure 1g), but they did not show whether this extends to Lrr1.

9) Unlike p97i, immunodepletion of the Npl4-Ufd1 complex delays but does not prevent CMG unloading (Figure 2a,b). Could this suggest that other p97 co-factors can compensate for CMG disassembly in the absence of Npl4-Ufd1?

10) The input blots for Npl4, Ufd1 and FLAG-rUbxn7 in Figure 3b and the Mcm7 blots in Extended Data Figure 5b should be replaced with longer exposures.

11) Why is FLAG-rUbxn7 largely deficient for binding to K48 Ub(2-7) in the presence of p97, Npl4 and Ufd1 in vitro (Extended Data Figure 4c)?

Reviewer #2 (Remarks to the Author):

In this paper, Kochenova et al. identify Ubxn7/UBXD7 as the p97 cofactor required for CMG disassembly together with Ufd1 and Npl4 (in *Xenopus* and mammalian cells). While characterising the role of the individual Ubxn7 domains in this process, they find that p97-Ufd1/Npl4 and Ubxn7 cooperatively assemble onto the ubiquitin chains. They show that FAF1 partially compensates for the absence of Ubxn7. They also demonstrate that the cofactor choice depends on an elusive DUB and on the length of the ubiquitin chain on MCM.

Overall, the paper presents several interesting findings that go beyond the process of CMG disassembly and are important for the ubiquitin field. For example, the data suggest that the ubiquitin-dependent assembly is widespread. However, a few points require clarification.

1) The finding that assembly depends on ubiquitin chains (data in figure 3) is interesting. The ubiquitin-dependent interaction between p97-Ufd1/Npl4 and Ubxn7 is maintained even in the presence of the Cullin inhibitor, clearly indicating that other ub ligases might be involved. This is also consistent with extended figure 1G, when the authors show that Ubxn7 UIM* can still form a complex with p97 - Ufd1/Npl4. However, I find surprising that the UBA mutant is still able to coIP p97-Ufd1/Npl4. How do the authors interpret this? Shouldn't E1 inhibition (Figure 3) and UBA mutant (extended Fig 1) give a similar result? (In this setting, even the UBA*/UIM* has a less dramatic effect, compare Figure 1a with Extended 1g)

2) The authors report FAF1 as being a back-up cofactor. The CMG unloading defect is increased by depleting both Ubxn7 and FAF1, but the process is not completely prevented. I acknowledge the authors discussing this. However, the blots seem to come from two different gels (Figure 4a and extended 5b). It is important to show all the lanes on the same membrane (as done in the other experiments). Considering that there is residual CMG unloading (lanes 10-12 in 4a), the authors should show that the stronger ub-MCM/CMG signals are not just an effect of longer exposure on the right hand-side gel.

Minor suggestions:

- In the text, line 110, the authors state that "surprisingly, Ubxn7UBA* and Ubxn7 lacking the entire UBA domain (Ubxn7 Δ UBA) still promoted substantial CMG unloading and p97 recruitment. But on lane

120 they write: "Like Ubxn7 Δ UBA, Ubxn7UIM* failed to completely unload CMG". The wording is confusing. I suggest the interpretation should either go towards "they partially rescue" or towards "the rescue is incomplete" for both mutant constructs (and in both sentences).

- The interaction FAF1-p97 UN is lost with Culi (Fig 3C). The authors write "Given that Faf1 co-IPs with CRL1 and CRL3 complexes, this suggests that p97/Ufd1-Npl4-Faf1 might assemble on ubiquitin chains made by these E3 ligases". Could the authors test this hypothesis by, e.g., depleting Cul2 from the extract (instead of Culi)?
- Extended Figure 3b: please show the bait blot (GST-Ufd1)

Reviewer #3 (Remarks to the Author):

This manuscript describes the assembly of a complex composed of the p97 ATPase, its cofactors and its target involved in extracting polyubiquitylated CMG from DNA during replication termination. While the co-factors Ufd1-Npl4 are known to recruit polyubiquitylated target proteins to p97, the regulatory UBA-UBX domain protein involved in CMG disassembly in vertebrates was not known. In a previous study, the authors found that in the presence of a p97 inhibitor, not only the CMG, but also p97, Ufd1, Npl4, and the UBA-UBX protein Ubxn7, are enriched on the chromatin during replication termination. They now show that Ubxn7 is required for recruitment of p97-Ufd1-Npl4 to ubiquitylated CMG, and promotes CMG unloading. Conversely, p97-Ufd1-Npl4 is also required for a stable interaction of Ubxn7 with the ubiquitylated CMG. Surprisingly, the interdependent complex between p97-Ufd1-Npl4 and Ubxn7 is not necessarily mediated by a ubiquitylated target but can also be formed on free ubiquitin chains. Furthermore, in absence of Ubxn7, another UBA-UBX protein, Faf1, acts partially redundantly in CMG unloading. While Faf1 seems to be less efficient in promoting CMG unloading, a Faf1 chimera containing Ubxn7's UIM domain enhanced this activity indicating the importance of this domain for optimal CMG unloading. Finally, the authors show that depletion of Ubxn7 from mammalian cells inhibits CMG unloading at the end of S-phase further confirming the important role of this protein in replication termination.

The p97 ATPase is an important and versatile protein involved in extracting proteins from various cellular structures. To allow specificity for the right substrate at the right time it interacts with many co-factors. How complex formation between p97, its cofactors and targets is regulated is poorly understood. This manuscript describes several important new insights into how such a complex is assembled at polyubiquitylated CMG to promote efficient CMG unloading after DNA replication. This is important to understand replication termination but also provides details on how other proteins are extracted by the p97 ATPase. Therefore, in my opinion this work is highly suitable for publication in Nature Communications. This study is very elaborate, well executed and describes all of the experiments required to draw the conclusions. I only have one minor comment that could be addressed.

The authors show how Faf1+UIM added in excess rescues CMG unloading (ED Figure 5b). However, p97 was not detected on the chromatin in the presence of p97i (ED Figure 6b). This is strange because p97 is detected abundantly on the chromatin in presence of Ubxn7 (+p97i) and Faf1+UIM should be assembled in the complex with a similar mechanism.

This can in part be explained by a shortening of the ubiquitin chain during the plasmid pull down. Indeed, in absence of Ubxn7 shorter chains appear and p97 recruitment is lost. To counteract this chain shortening the authors add Ub-V5 inhibiting deubiquitylation. This rescues p97 recruitment in the presence of Faf1+UIM to a small extent. However, it is not clear from the figure (Figure 4e) whether this also rescues the ubiquitin chain length since the gel seems to have less resolution

compared to the other experiments (for example in Figure 1a, ED 6a, b, and c) and it is not quantified as in Figure ED 6d. Therefore, we can't be sure whether the rescue in p97 recruitment is due to the ubiquitin chain lengthening.

It would be great if the authors can provide an explanation why the recruitment of p97 is so different in the case of Ubxn7 compared to Faf1+UIM since it seems this is only to a minor extent caused by ubiquitin chain shortening.

REVIEWER COMMENTS

We thank the reviewers for their detailed, constructive comments. Although we addressed most of the comments and added new data, some comments would have required extensive experimentation that would not significantly alter the conceptual message of the paper. Performing high quality structure-function analysis in egg extracts is very time-consuming, and the work presented already comprises 6 years of hard work by a post-doc and an additional year by a student. In addition, overlapping papers were recently published by the Labib and Gambus groups. Given these considerations, we feel that delaying the work further is not productive. We hope the reviewers agree that the experiments presented represent a compelling package that makes conceptually novel contributions to our understanding of how p97 acts on its clients in a physiological setting. The reviews are reproduced in their entirety, our responses are in green.

Reviewer #1 (Remarks to the Author):

The study by Kochenova et al. investigates how the p97 ATPase disassembles the replicative CMG helicase upon its ubiquitination during DNA replication termination. Using *Xenopus* egg extracts, they found that the p97 co-factor Ubxn7 is required for efficient CMG unloading in a manner that depends on its ability to interact with ubiquitin chains, neddylated CRL complexes and p97 via the UBA, UIM and UBX domains, respectively. They show that recruitment of Ubxn7 and p97/Npl4-Ufd1 to CMG is interdependent, and this cooperative assembly involves interactions between Ubxn7, p97, Ufd1-Npl4 and ubiquitin, but not a p97 substrate per se, a mechanism that seems likely to apply more broadly to interactions between p97-co-factor complexes and their ubiquitinated targets. The authors also provide evidence to suggest that Faf1, another p97 UBA-UBX co-factor, contributes to CMG unloading in the absence of Ubxn7, and that Ubxn7 is also important for CMG disassembly in human cells.

The new findings showing cooperative assembly of p97 and its co-factors on ubiquitin chains involving multivalent interactions are potentially important, for the most part convincing and should be of broad interest. I therefore believe the manuscript is appropriate in principle for Nature Communications. However, there are several aspects of the study that need to be strengthened significantly before publication in this journal can be recommended. Specifically, the following points should be addressed:

1) The inability of Npl4 LV-Ufd1 deltaUT3 to support CMG disassembly in Npl4-depleted extracts (Figure 2) is not convincing. First, Extended Data Figure 2a,c indicate that much less recombinant Ufd1 deltaUT3 mutant was present relative to the WT protein.

Ufd1^{ΔUT3} only appears to be underloaded due to poor recognition of this construct by polyclonal antibodies raised to the full-length *Xenopus* Ufd1 protein (Figure R1, compare a and b). Indeed, based on an antigenicity algorithm (<http://imed.med.ucm.es/Tools/antigenic.pl>), the most antigenic portion of the protein was lost in the Ufd1^{ΔUT3} truncation (not shown), which explains the poor detection of this construct.

The data in the paper further documents that we added equivalent amount of Ufd1^{ΔUT3}-Npl4 as the other constructs: For purification of all Ufd1-Npl4 mutant variants, we use GST-tagged Ufd1 and glutathione affinity

Figure R1: Ufd1 antibody recognizes Ufd1^{ΔUT3} poorly. (a) Recombinant wild-type and mutant variants of Ufd1-Npl4 analyzed by SDS PAGE and Coomassie staining. (b) Western blot analysis of the Ufd1-Npl4 preparations shown in (a) using the indicated antibodies. Note that despite equivalent loading in (a), Ufd1^{ΔUT3} is poorly recognized in (b).

purification. This strategy recovers Npl4 only in complex with Ufd1; the fact that equivalent levels of Npl4 are recovered with Ufd1^{ΔUT3} as the other Ufd1 constructs (Extended Data Figure 2a and c) shows that equivalent concentration of all constructs were present. Furthermore, recombinant Ufd1^{ΔUT3}-Npl4 with a GST tag on Ufd1 (GST-Ufd1^{ΔUT3}) recovers similar amount of p97 as the wild-type protein from egg extracts (Extended data Fig. 2d, lanes 9-10, 14-15), further confirming that all recombinant Ufd1-Npl4 variants are added at similar concentrations to extracts.

Second, the deltaUT3 mutation deletes more than half of the Ufd1 protein, which might impair Npl4-Ufd1 function in ways that are not solely due to loss of ubiquitin binding. The authors could attempt to generate and test a Ufd1 mutant that abrogates ubiquitin binding without removing a large portion of the protein.

To address this criticism, we used AlphaFold multimer to generate a structure of *Xenopus* Ufd1-Npl4. The structure shows that the UT6 domain (Figure R2, lime green), which binds Npl4 (blue), and the UT3 domain (forest green), which binds ubiquitin, do not interact. Thus, removal of the UT3 domain (deletion of Met1 – Asp215 in Figure R2) is unlikely to impact Npl4-UT6. Indeed, GST-Ufd1^{ΔUT3} retains the ability to bind p97 and Npl4 via its SHP1 and NBM motifs within the UT6 domain (Extended Data Fig. 2d, lane 10). Moreover, Ufd1^{ΔUT3} is a commonly used Ufd1 variant for biochemical studies (Bodnar et al., 2018; Ye et al., JCB, 2003; Hetzer et al., Nat Cell Biol, 2001), and there are no well characterized point mutations in the UT3 domain.

Figure R2: AlphaFold multimer structure of *Xenopus laevis* Ufd1-Npl4 heterodimer. The UT3 and UT6 domains of Ufd1 are colored in forest green and lime, respectively.

2) Based on corresponding point mutations in yeast Npl4, the authors state that the xNpl4 AAE triple point mutant is expected to lose ubiquitin unfolding activity (Figure 2). More direct evidence for this claim should be provided.

The cryo-EM structure of the human p97^{UFD1-NPL4} complex with a model polyubiquitylated substrate (Pan et al., NSMB, 2021) is highly related to yeast Cdc48^{Ufd1-Npl4}, implying a conserved role for the Npl4 groove in initiation of ubiquitin chain unfolding (Twomey et al., Science, 2019). Moreover, the three residues we mutated, as well as their positions in the ubiquitin-binding channel of Npl4, are highly conserved from yeast to humans (Figures R3, R4). Since the equivalent yeast mutant has been characterized extensively (Twomey et al., Science, 2019; Ji et al., Mol Cell, 2022), we believe that performing the unfolding assays with the *Xenopus* protein would not add conceptually to the study while involving a large amount of work. We therefore elected not to perform these experiments.

Figure R3. The alignment of Npl4 amino acid sequences from different organisms. The three mutated residues in the MPN domain are indicated with asterisks.

3) It would be informative to probe for Faf1 in the experiments in Figure 1a and 2a,c.

We have now probed for Faf1 in Ubxn7- and Ubxn7/Npl4-depleted extract (new Fig. 4e). However, we were not able to observe reproducible changes in the endogenous Faf1 binding to chromatin in this setting due to a variable background level of Faf1 recovery (Fig. 4e, compare “No DNA” lane and lanes 1-3).

4) The data in Figure 4 are somewhat underwhelming. The authors generated a Faf1 chimera containing an insertion of the Ubxn7 UIM motif to test the idea that Ubxn7 is more active for CMG unloading due to CRL2/Lrr1 interaction via the UIM. However, they did not provide evidence that this actually enables Faf1 to interact with CRL2/Lrr1. The authors should also test whether the UIM insertion impacts Faf1’s ability to interact with p97, a possibility they discuss on page 16.

As requested, we IP’d Faf1^{UIM} from NPE and showed that it recovers more neddylated Cul2 than the wild-type protein while not affecting p97 binding (new Extended Data Fig.6b, lanes 15 and 16).

The ability of the Faf1/UIM chimera to support p97/Npl4-Ufd1 recruitment in the presence of Ub-VS (Figure 4e) is so weak that it seems questionable this has any real significance. In general, the notion that Faf1/UIM is less active than Ubxn7 in unloading CMG through failure to restrain a deubiquitinating enzyme acting on Mcm7 does not seem well supported by the presented data.

We agree that the original Ub-VS data was not very compelling, and we have removed it (former Figure 4e), especially since we now improved low level detection of p97 and Npl4 by using more sensitive ECL reagents. This analysis shows more clearly that Faf1^{UIM} is better than Faf1^{WT} in recruiting p97 (new Fig. 4e). Nevertheless, compared to Ubxn7, Faf1^{UIM} still binds poorly to chromatin and recruits low levels of p97^{Ufd1-Npl4}, despite showing substantial CMG unloading activity. p97 recruitment by Faf1 was not significantly improved by increasing the size of the UIM fragment inserted into Faf1, moving the location of the UIM insertion, replacing Faf1’s UB domain with Ubxn7’s UB domain, or lowering the stringency of washing buffers (data not shown). Future work will be required to explain why such low levels of p97 are recovered in the presence of Faf1.

5) Does Faf1’s UBA and UB domains promote cooperative assembly of p97/Ufd1-Npl4-Faf1 on ubiquitin chains, similar to Ubxn7?

Previous work (Song et al., Mol Cell Biol, 2005; Lee et al., JBC, 2013) has shown that Faf1 interacts with p97 via FAF1’s UB domain and with ubiquitin chains via its UBA domain. As we showed in Fig. 3c and Extended Data Fig. 2d, Faf1’s binding to p97^{Ufd1-Npl4} requires ubiquitin chains. These results strongly argue that the assembly of p97-Ufd1-Npl4-Faf1 complex relies on Faf1’s ability to interact with ubiquitin

Figure R4. MPN Structures. (a) The MPN domain structure of *Chaetomium thermophilum* Npl4 (PDB: 6cdd) colored according to the degree of conservation of amino acid residues, as determined by the ConSurf Server. The location of the LWR residues mutated to AAE are labeled with yellow dots. (b) The alignment of MPN domains of *Xenopus laevis* (beige) and *Saccharomyces cerevisiae* (blue) generated using AlphaFold. The mutated LWR (IWR in yeast) residues within the MPN groove are labeled with black dots.

chains and p97. However, performing the mutational analysis of Faf1 would require redoing all the experiments, which is many months of work, while adding very little conceptually. We therefore elected not to perform these experiments.

6) The examination of Faf1's role in CMG unloading should be extended by analyzing the impact of FAF1 depletion in Wild-type and UBXD7 KO cells (Figure 5a,b). It would also be relevant to test the ability of WT UBXD7 and UBA and/or UBX mutants to rescue the CMG disassembly defect of UBXD7 KO cells via stable reconstitution, which should be well feasible given that UBXD7 was knocked out in a Flp-in background.

Unfortunately, FAF1 knockdown cells do not release from thymidine block, and a significant fraction of cells remain at the G1/S transition (Figure R5), so we were not able to look at the role of hFAF1 in CMG unloading at the end of S phase in mammalian cells.

Figure R5: FAF1 knock-down prevents release from cell cycle arrest

Cell cycle analysis of WT and UBXD7 knockout HFT cells treated with control or FAF1 siRNA. DNA content of asynchronous cells, double thymidine arrested cells (0) and cells released from arrest for 8 hours (8) are shown.

Regarding the mutational analysis of Ubxn7 in human cells that was requested, this adds a large amount of work without adding any new concepts. We therefore elected not to perform these experiments.

Minor points:

7) Does the UBX* mutation affect Ubxn7 interaction with K48-Ub(2) in vitro, as tested for the UBA* mutant in Extended Data Figure 1f?

In contrast to the UBA* mutation, UBX* does not affect Ubxn7's ability to bind K48-Ub2 in vitro (new Extended Data Fig. 1f, lane 8e).

8) The authors found that Ubxn7 binds Cul2 in a UIM-dependent manner (Extended Data Figure 1g), but they did not show whether this extends to Lrr1.

Unfortunately, when we performed the relevant Ubxn7 pull-down and blotted for Lrr1, a background band appeared at the position of Lrr1. Therefore, this experiment is not feasible with our current reagents. We have pointed out this caveat in the revised manuscript in the relevant Extended Data Fig. 1g legend.

9) Unlike p97i, immunodepletion of the Npl4-Ufd1 complex delays but does not prevent CMG unloading (Figure 2a,b). Could this suggest that other p97 co-factors can compensate for CMG disassembly in the absence of Npl4-Ufd1?

While we cannot rule out a minor contribution of other known or unknown p97 co-factors in the residual CMG unloading, we favor the idea that it results from incomplete Ufd1-Npl4 depletion. Nevertheless, we now acknowledge both possibilities more explicitly in the text.

10) The input blots for Npl4, Ufd1 and FLAG-rUbx7 in Figure 3b and the Mcm7 blots in Extended Data Figure 5b should be replaced with longer exposures.

We have done this.

11) Why is FLAG-rUbx7 largely deficient for binding to K48 Ub(2-7) in the presence of p97, Npl4 and Ufd1 in vitro (Extended Data Figure 4c)?

Recombinant Ubx7 doesn't efficiently bind ubiquitin chains in vitro with or without p97^{Ufd1-Npl4}, unless ubiquitin chains and Ubx7 are added at a very high concentration (80 μM and 2 μM, respectively, as in Fig. 4c vs. 0.8 μM and 10 nM in Extended Data Figure 4c). Since p97^{Ufd1-Npl4} readily binds Ubx7 in the absence of ubiquitin chains in reconstitution reactions, in Extended Data Figure 4c we were attempting to detect cooperative assembly of the complex on ubiquitin chains by lowering the concentrations of all components.

Reviewer #2 (Remarks to the Author):

In this paper, Kochenova et al. identify Ubx7/UBXD7 as the p97 cofactor required for CMG disassembly together with Ufd1 and Npl4 (in *Xenopus* and mammalian cells). While characterising the role of the individual Ubx7 domains in this process, they find that p97-Ufd1/Npl4 and Ubx7 cooperatively assemble onto the ubiquitin chains. They show that FAF1 partially compensates for the absence of Ubx7. They also demonstrate that the cofactor choice depends on an elusive DUB and on the length of the ubiquitin chain on MCM.

Overall, the paper presents several interesting findings that go beyond the process of CMG disassembly and are important for the ubiquitin field. For example, the data suggest that the ubiquitin-dependent assembly is widespread. However, a few points require clarification.

1) The finding that assembly depends on ubiquitin chains (data in figure 3) is interesting. The ubiquitin-dependent interaction between p97-Ufd1/Npl4 and Ubx7 is maintained even in the presence of the Cullin inhibitor, clearly indicating that other ub ligases might be involved. This is also consistent with extended figure 1G, when the authors show that Ubx7 UIM* can still form a complex with p97 - Ufd1/Npl4. However, I find surprising that the UBA mutant is still able to coIP p97-Ufd1/Npl4. How do the authors interpret this? Shouldn't E1 inhibition (Figure 3) and UBA mutant (extended Fig 1) give a similar result? (In this setting, even the UBA*/UIM* has a less dramatic effect, compare Figure 1a with Extended 1g).

Although we verified that these point mutations affect di-ubiquitin binding of the purified UBA domain (former Extended data Fig. 1f) and full-length Ubx7 (new Extended data Fig. 1f, lane 7), it appears that in extracts, UBA* still exhibits quite efficient binding to polyubiquitin chains (Extended data figure 1g, lane 39). In the new Extended Data Fig. 3b (lanes 10, 12, 14), we now show that full deletion of the UBA domain greatly reduces, and the combination of a full UBA deletion with mutation in the UIM motif effectively eliminates binding to p97^{Ufd1-Npl4}. Thus, complete loss of Ubx7 interaction with p97^{Ufd1-Npl4} requires complete abrogation of its CRL2 and ubiquitin binding activities.

2) The authors report FAF1 as being a back-up cofactor. The CMG unloading defect is increased by depleting both Ubx7 and FAF1, but the process is not completely prevented. I acknowledge the authors discussing this. However, the blots seem to come from two different gels (Figure 4a and extended 5b). It is important to show all the lanes on the same membrane (as done in the other experiments).

Considering that there is residual CMG unloading (lanes 10-12 in 4a), the authors should show that the

stronger ub-MCM/CMG signals are not just an effect of longer exposure on the right hand-side gel. While the samples for this experiment were indeed resolved on different gels due to an inability to fit all of them on one gel, the two membranes for each indicated protein were processed and imaged in parallel under identical conditions. Images from the same exposure (see an example of raw images for Mcm7 in Figure R6) were used for lanes 1-9 and 10-21 for Fig. 4b and Extended data Fig. 5b). Uncropped images for other proteins are provided as Source data.

Minor suggestions:

- In the text, line 110, the authors state that “surprisingly, Ubxn7UBA* and Ubxn7 lacking the entire UBA domain (Ubxn7 Δ UBA) still promoted substantial CMG unloading and p97 recruitment. But on lane 120 they write: “Like Ubxn7 Δ UBA, Ubxn7UIM* failed to completely unload CMG”. The wording is confusing. I suggest the interpretation should either go towards “they partially rescue” or towards “the rescue is incomplete” for both mutant constructs (and in both sentences).

We thank reviewer for this suggestion, which we have implemented.

Figure R6. Uncropped Mcm7 blots for Fig. 4a (left) and Extended Data Fig. 5b (right).

- The interaction FAF1-p97 UN is lost with Culi (Fig 3C). The authors write “Given that Faf1 co-IPs with CRL1 and CRL3 complexes, this suggests that p97/Ufd1-Npl4-Faf1 might assemble on ubiquitin chains made by these E3 ligases”. Could the authors test this hypothesis by, e.g., depleting Cul2 from the extract (instead of Culi)?

Unfortunately, we have tried and failed to generate depletion-quality antibodies for Cul2.

- Extended Figure 3b: please show the bait blot (GST-Ufd1)

We have now replaced Extended Data Fig. 3b with a new replicate showing the GST (bait) western blot.

Reviewer #3 (Remarks to the Author):

This manuscript describes the assembly of a complex composed of the p97 ATPase, its cofactors and its target involved in extracting polyubiquitylated CMG from DNA during replication termination. While the co-factors Ufd1-Npl4 are known to recruit polyubiquitylated target proteins to p97, the regulatory UBA-UBX domain protein involved in CMG disassembly in vertebrates was not known. In a previous study, the authors found that in the presence of a p97 inhibitor, not only the CMG, but also p97, Ufd1, Npl4, and the UBA-UBX protein Ubxn7, are enriched on the chromatin during replication termination. They now show that Ubxn7 is required for recruitment of p97-Ufd1-Npl4 to ubiquitylated CMG, and promotes CMG unloading. Conversely, p97-Ufd1-Npl4 is also required for a stable interaction of Ubxn7 with the ubiquitylated CMG. Surprisingly, the interdependent complex between p97-Ufd1-Npl4 and Ubxn7 is not necessarily mediated by a ubiquitylated target but can also be formed on free ubiquitin chains. Furthermore, in absence of Ubxn7, another UBA-UBX protein, Faf1, acts partially redundantly in CMG unloading. While Faf1 seems to be less efficient in promoting CMG unloading, a Faf1 chimera

containing Ubxn7's UIM domain enhanced this activity indicating the importance of this domain for optimal CMG unloading. Finally, the authors show that depletion of Ubxn7 from mammalian cells inhibits CMG unloading at the end of S-phase further confirming the important role of this protein in replication termination.

The p97 ATPase is an important and versatile protein involved in extracting proteins from various cellular structures. To allow specificity for the right substrate at the right time it interacts with many co-factors. How complex formation between p97, its cofactors and targets is regulated is poorly understood. This manuscript describes several important new insights into how such a complex is assembled at polyubiquitylated CMG to promote efficient CMG unloading after DNA replication. This is important to understand replication termination but also provides details on how other proteins are extracted by the p97 ATPase. Therefore, in my opinion this work is highly suitable for publication in Nature Communications. This study is very elaborate, well executed and describes all of the experiments required to draw the conclusions. I only have one minor comment that could be addressed.

The authors show how Faf1+UIM added in excess rescues CMG unloading (ED Figure 5b). However, p97 was not detected on the chromatin in the presence of p97i (ED Figure 6b). This is strange because p97 is detected abundantly on the chromatin in presence of Ubxn7 (+p97i) and Faf1+UIM should be assembled in the complex with a similar mechanism.

This can in part be explained by a shortening of the ubiquitin chain during the plasmid pull down. Indeed, in absence of Ubxn7 shorter chains appear and p97 recruitment is lost. To counteract this chain shortening the authors add Ub-VS inhibiting deubiquitylation. This rescues p97 recruitment in the presence of Faf1+UIM to a small extent. However, it is not clear from the figure (Figure 4e) whether this also rescues the ubiquitin chain length since the gel seems to have less resolution compared to the other experiments (for example in Figure 1a, ED 6a, b, and c) and it is not quantified as in Figure ED 6d. Therefore, we can't be sure whether the rescue in p97 recruitment is due to the ubiquitin chain lengthening.

It would be great if the authors can provide an explanation why the recruitment of p97 is so different in the case of Ubxn7 compared to Faf1+UIM since it seems this is only to a minor extent caused by ubiquitin chain shortening.

We have removed the Ub-VS data (former Figure 4e), as we now significantly improved the detection of low levels of p97 and Npl4 by using more sensitive ECL reagents (new Fig. 4e). We tested four different ways of increasing p97 recruitment by the Faf1^{UIM} chimera (increasing the size of the UIM fragment inserted into Faf1, moving the location of the UIM insertion, replacing Faf1's UBX domain with Ubxn7's UBX domain, or lowering the stringency of washing buffers), but in each case, the result was negative (data not shown). We now briefly describe these experiments in the revised manuscript, which will inform any future attempts to explain the poor p97 recruitment by Faf1.

REVIEWERS' COMMENTS

Reviewer #1 (Remarks to the Author):

As mentioned in my original referee report, the study by Kochenova et al. provides important new insights into how VCP/p97 acts on client proteins via cooperative assembly with its co-factors on ubiquitin chains involving multivalent interactions, which should be of broad interest. While the authors did not address all my concerns, the key conclusions are overall convincing, and given that other recent papers have reported overlapping findings I believe the current study is timely and should be published soon. Altogether, I therefore support publication of the revised manuscript in Nature Communications.

Reviewer #2 (Remarks to the Author):

The authors have addressed all my concerns, and I am now happy to support this manuscript for publication.

Reviewer #3 (Remarks to the Author):

The revised version of the manuscript "Cooperative assembly of p97 complexes involved in replication termination" contains several new experiments and extended explanations for some of the issues raised by the reviewers.

My comment was well addressed as Figure 4e was replaced by a new experiment that showed the differences in ubiquitin chain length much better. Furthermore, the authors explained all their attempts to further promote p97 recruitment by the Faf1UIM chimera. While this has not been successful it should no longer delay the publication of this manuscript. This is a very extensive study with several novel insights that deserves to be published.